# Selecting Samples on Graphs: A Unified Dataset Pruning Framework for Lossless Training Acceleration

**Dongyue Wu** [1 2]  **Zilin Guo** [1]  **Xiaoyu Li** [2]  **Jiajia Liu** [2]  **Jingdong Chen** [2]  **Nong Sang** [1]  **Changxin Gao** [1]

## Abstract

The rapid growth of modern training datasets has significantly increased computational cost, motivating dataset pruning (DP) methods which retain only a subset of informative samples to reduce training cost. Existing pruning criteria typically rely on either intrinsic signals that assess samples independently or extrinsic signals that promote diversity via pairwise relations. While effective in their own specific regimes, each captures only one aspect of sample utility and lacks robustness across different pruning ratios or data distribution. In this work, we present a unified graph-based DP framework. By modeling the dataset as a weighted graph, where node weights encode intrinsic value and edge weights encode extrinsic value, DP can be cast as a Maximum Weight Clique Problem (MWCP). Although MWCP is NP-hard, its structure admits a principled greedy solution based on sample-wise marginal gains. Under a few mild conditions, we further prove that this unified objective enjoys a formal approximation guarantee, which applies to a broad family of importance metrics and provides practical design guidelines. Extensive experiments show that our method outperforms existing DP methods while substantially reducing training cost, reducing training time by over 40% without sacrificing accuracy on ImageNet-1k with ResNet-50.

## 1. Introduction

Deep learning on large datasets has driven breakthroughs across language, vision, and multimodal reasoning, fueling the rise of LLMs (Achiam et al., 2023), CLIP-based vision–language models (Radford et al., 2021), diffusion generators (Rombach et al., 2022), and powerful downstream models such as SAM (Kirillov et al., 2023). Yet the benefits of numerous data come at substantial computational cost: training modern models often requires millions to billions of examples and may span weeks or months, even on large GPU clusters. As datasets continue to scale continuously, the data efficiency of training has emerged as the bottleneck.

A growing line of work seeks to reduce dataset size without degrading downstream performance. These approaches fall into two families: dataset distillation, which synthesizes a compact set of virtual samples, and dataset pruning, which selects an informative subset of real data. Dataset distillation(Zhao & Bilen, 2023; Cazenavette et al., 2022; Wang et al., 2022; Nguyen et al., 2020; Zhao et al., 2020) can be powerful at extreme compression ratios, but synthetic examples may drift from the natural data manifold, limiting interpretability and raising safety concerns. Dataset pruning(Paul et al., 2021; Yang et al., 2022; Toneva et al., 2018b; Hong et al., 2024), by contrast, evaluates sample-wise importance and selects informative real samples, preserving the natural data and offering more interpretable pruning decisions that better align with practical deployment needs.

Most pruning methods follow a similar pipeline: compute an importance score for each training example, then retain the highest-scoring subset. Thus, their effectiveness is determined primarily by how importance is defined. Existing criteria can be broadly categorized into *Intrinsic* and *Extrinsic* formulations. Intrinsic criteria evaluate samples independently using instance-level signals—such as loss values (Kawaguchi & Lu, 2020; Qin et al., 2023; Paul et al., 2021; Mindermann et al., 2022; Jiang et al., 2019; Loshchilov & Hutter, 2015), gradients (Killamsetty et al., 2021a; Koh & Liang, 2017; Tan et al., 2024; Yang et al., 2022; Hong et al., 2024), forgetting scores (Toneva et al., 2018b), or uncertainty estimates (Coleman et al., 2019; He et al., 2024). Extrinsic criteria instead evaluate each sample through its relationships with others, promoting diversity and coverage via objectives such as K-center (Sener & Savarese, 2018), herding (Chen et al., 2012), or clustering-based selection (Zheng et al., 2023; Sorscher et al., 2022).

---

[1]State Key Laboratory of Multispectral Information Intelligent Processing Technology, School of Artificial Intelligence and Automation, Huazhong University of Science and Technology, Wuhan, China [2]Ant Group, Hangzhou, China. Correspondence to: Changxin Gao <cgao@hust.edu.cn>.

While both directions are effective in the regimes they target, intrinsic criteria emphasize per-sample informativeness but ignore redundancy, whereas extrinsic formulations encourage global diversity yet overlook how informative each sample is individually. This dichotomy suggests that sample importance is inherently multi-faceted and cannot be fully captured by either perspective alone.

These observations indicate that the value of a sample should be determined jointly by its intrinsic learning potential and its extrinsic interactions with other samples. A few works (Maharana et al., 2024; Tan et al., 2025) have taken steps in this direction by combining multiple criteria, but they ultimately rely on a *fixed* instantiation of sample importance. Such formulations obscure the underlying structure of dataset pruning as a subset selection problem and limit adaptability across different pruning scenarios, such as pruning ratios or datasets. Accordingly, this calls for a more principled framework that models both aspects within a unified objective, rather than prescribing a single heuristic metric. Moreover, such a framework should admit efficient optimization and retain theoretical guarantees even when the intrinsic and extrinsic components are flexibly instantiated.

To address these challenges, we introduce **UGIES**, a **U**nified **G**raph-based **I**mportance **E**valuation **S**ystem for dataset pruning. UGIES models the dataset as a graph whose node weights encode intrinsic importance and whose edge weights encode pairwise extrinsic interactions. Selecting an informative and non-redundant subset then naturally corresponds to finding a high-quality subgraph, which can be formally cast as a Maximum-Weight Clique Problem (MWCP). However, the MWCP is NP-hard, yet it exhibits a favorable property that enables efficient approximation. When pruning only a single node, the optimal decision is determined by marginal gains. Leveraging this observation, we derive a principled greedy strategy that evaluates samples through sample-wise marginal importance, yielding an efficient and scalable pruning algorithm rather than a heuristic rule. We prove that under a few mild and interpretable conditions, the unified objective becomes submodular, enabling an efficient greedy solver with a theoretically grounded performance guarantee. Extensive experiments demonstrate that the proposed framework consistently outperforms existing methods. Our contributions are summarized as threefold:

1. We cast dataset pruning from a graph view as the classical MWCP, based on which we introduce a unified pruning framework that measures both intrinsic and extrinsic values of samples.

2. Although the MWCP is NP-hard, we exploit its structural properties and design a principled greedy strategy that efficiently approximates the optimal solution.

3. We prove that a broad family of importance metrics satisfying a few mild and interpretable conditions admits

formal approximation guarantees. These conditions serve as principled guidelines for designing importance metrics with theoretical support.

## 2. Related Work

We group prior pruning works into two complementary families: intrinsic and extrinsic methods.

**Intrinsic criteria.** Intrinsic methods estimate per-sample learnability from individual signals such as loss, prediction behavior, uncertainty, or gradients. SB (Jiang et al., 2019) and InfoBatch (Qin et al., 2023) prioritize high-loss samples, while EL2N (Paul et al., 2021) measures sample difficulty using the average prediction error across epochs. Uncertainty-based approaches, including SVP (Coleman et al., 2019) and Dyn-Unc (He et al., 2024), select samples with high predictive entropy, and trajectory-based indicators such as Forgetting (Toneva et al., 2018b) track transitions in correctness over training. Optimization-oriented formulations further assess importance via influence functions (Koh & Liang, 2017), Hessian–gradient interactions (Yang et al., 2022), or gradient alignment (Tan et al., 2024; Killamsetty et al., 2021a). Despite their efficiency and interpretability, intrinsic methods treat samples independently and therefore struggle to control redundancy under aggressive pruning.

**Extrinsic criteria.** Extrinsic methods incorporate interactions among samples to preserve geometry, diversity, or distributional coverage. Mean-embedding matching (Chen et al., 2012) and k-center selection (Sener & Savarese, 2018) encourage representative coverage of feature space, while clustering-based approaches (Zheng et al., 2023; Sorscher et al., 2022) identify exemplars via grouping in learned representations. DivBS (Hong et al., 2024) maintains local diversity by removing samples with redundant gradient directions, and density-aware strategies such as PFB (Wu et al., 2025) favor points in low-density regions of the dataset. These approaches improve coverage but may overlook per-sample learnability and often involve heuristic or combinatorial objectives lacking principled optimization guarantees.

**Recent hybrid methods.** Early active learning works like (Senzaki & Hamelain, 2023) also propose to combine sample-independent value with relational importance for a comprehensiveness. Recent pruning approaches also attempt to combine intrinsic and relational information. $\mathcal{D}^2$-pruning (Maharana et al., 2024) propagates difficulty scores on a graph, and InfoMax (Tan et al., 2025) integrates instance-level and pairwise signals via an information-theoretic potential. Although effective, these methods typically rely on a *fixed heuristic metric*. By contrast, our method introduces the unified and principled objective that accommodates a large family of intrinsic and extrinsic metrics. We provide flexible choices of metrics rather than a

fixed measurement, but still manage to make a family of objectives submodular under mild conditions. This yields general design guidelines, a theoretically grounded greedy solver, and the flexibility to accommodate diverse intrinsic and extrinsic metrics across datasets and pruning regimes.

**Submodularity-based data selection.** Prior methods such as CRAIG (Mirzasoleiman et al., 2020), GLISTER (Killamsetty et al., 2021b), and GRAD-MATCH (Killamsetty et al., 2021a) also exploit submodularity to enable greedy data selection with theoretical guarantees. However, they usually derive submodularity merely for their own specific objective function. UGIES instead treats submodularity as a design principle for a unified intrinsic-extrinsic importance family, where different metrics can be instantiated while preserving the submodularity and greedy optimizability under mild conditions.

# 3. Methodology

## 3.1. Problem Formulation

Given the full training dataset $\mathcal{T} = \{x_i\}_{i=1}^N$ containing $N$ samples, our goal is to select a compact yet representative subset $\mathcal{S} \subseteq \mathcal{T}$ with a cardinality $|\mathcal{S}| = (1-p)N$, where $p$ denotes the desired pruning ratio. The selected subset $\mathcal{S}$ is then used to train the target model. By discarding the $p \cdot N$ samples, dataset pruning reduces the training cost and enables practical acceleration. Beyond computational savings, the selected subset is expected to preserve or ideally even improve the performance and generalization achieved by the full dataset training. Therefore, $\mathcal{S}$ should consist of the most valuable samples that collectively contribute to learning. Thus, dataset pruning can be formulated as a subset selection problem:

$$\max_{\mathcal{S} \subseteq \mathcal{T}} f(\mathcal{S}) = \sum_{x_i \in \mathcal{S}} \mathcal{I}(x_i|\mathcal{S}), \quad \text{s.t.} \ |\mathcal{S}| = (1-p)|N|, \quad (1)$$

where $\mathcal{I}(x_i|\mathcal{S})$ denotes an abstract importance valuation of sample $x_i$. $\mathcal{I}(x_i|\mathcal{S})$ is not restricted to be a purely instance-wise quantity, but may implicitly depend on the selected subset $\mathcal{S}$ through inter-sample interactions. This formulation serves as a conceptual starting point for designing our principled objectives.

## 3.2. Principles of Dataset Pruning

Here, we first identify the following principles that an ideal dataset pruning method should satisfy as the guidelines to design our own pruning objectives:

**(1) Comprehensiveness.** An effective pruning objective should capture both the inherent value of individual samples and their collective contribution in the context of the selected subset. Hence, the objective should reflect two complementary components:

- *Intrinsic importance* $\mathcal{I}^{\text{in}}(x_i)$ measures the inherent learning utility of a sample, such as its difficulty, uncertainty, or informativeness. It is an instance-wise property, independent of other samples.

- *Extrinsic importance* $\mathcal{I}^{\text{ex}}(x_i|\mathcal{S})$ characterizes the contribution of a sample through its interactions with other samples to control redundancy, diversity, or coverage.

**(2) Flexibility.** A unified importance formulation should accommodate diverse choices of intrinsic and extrinsic criteria and allow their relative influence to be flexibly adjusted. This flexibility is important for adapting to different pruning ratios, dataset characteristics, and learning scenarios.

These principles impose structural requirements on the pruning objective and will guide our formulation in the Sec.3.3.

## 3.3. Rethinking Pruning from a Graph Perspective

Motivated by the principles in Section 3.2, we seek a formulation that jointly accounts for the intrinsic learning utility of individual samples and their extrinsic contributions arising from interactions within a selected subset. In particular, the extrinsic importance introduced by redundancy control and diversity promotion is inherently interaction-based, depending on relationships between pairs of samples. As such individual value and pair-wise interactions are explicitly modeled, the dataset pruning objective admits a canonical representation as a weighted graph.

Given the full training dataset $\mathcal{T} = \{x_i\}_{i=1}^N$, we construct an undirected weighted graph $G = (V, E)$, where each vertex $v_i \in V$ corresponds to a sample $x_i$, and each edge $(v_i, v_j) \in E$ encodes their pair-wise interaction. Consistent with the intrinsic–extrinsic decomposition, we assign vertex weights and edge weights as follows

$$w_i = \alpha \mathcal{I}^{\text{in}}(x_i), \qquad a_{ij} = g(D(x_i, x_j)), \quad (2)$$

where $w_i$ captures the intrinsic importance of $x_i$, $\alpha$ balances intrinsic and extrinsic terms, $D(\cdot, \cdot)$ denotes a distance or similarity metric between samples (also accommodates DivBS and PFB, see Sec.D.3), and $g(\cdot)$ maps this metric to a desired scale. The edge weight $a_{ij}$ quantifies the collective contribution incurred when $x_i$ and $x_j$ are simultaneously kept in $\mathcal{S}$.

**Maximum Weight Clique Formulation.** Under this construction, selecting a subset of samples with maximal overall utility corresponds to selecting a subset of vertices whose induced subgraph maximizes the total vertex and edge weights. This naturally leads to the following classical combinatorial optimization problem in graph theory.

**Definition 3.1** (Maximum Weight Clique Problem)**.** Given an undirected graph $G = (V, E)$ with vertex weights $\{w_i\}_{v_i}^V$

and edge weights $\{a_{ij}\}_{(v_i,v_j)}^E$, a *clique* $C \subseteq V$ is a subset of vertices such that every pair of vertices in $C$ is connected by an edge. The *Maximum Weight Clique Problem* (MWCP) seeks a clique of fixed cardinality $b$ that maximizes

$$\max_{C \subseteq V} \left[ \sum_{v_i \in C} w_i + \sum_{\{v_i,v_j\} \subseteq C} a_{ij} \right], \qquad \text{s.t. } |C| = b. \quad (3)$$

*Remark* 3.2. Under the graph construction in Eq. (2), the MWCP objective in Eq. (3) is exactly equivalent to maximizing the combined intrinsic and extrinsic importance of a selected subset of samples. Vertex weights correspond to intrinsic importance, while edge weights encode pairwise extrinsic interactions. Thus, dataset pruning is exactly equivalent to solving an MWCP on the constructed graph.

**Sample-wise Reformulation.** For algorithmic development, it is convenient to rewrite the set form Eq. (3) into an equivalent sample-wise form that measures the contribution of each selected sample explicit. Moreover, this reformulation does not alter the underlying MWCP. Let $\mathcal{S} \subseteq \mathcal{T}$ denote the subset corresponding to a clique $C$. By grouping the vertex and edge terms in Eq. (3) by samples, we can obtain

$$\max_{\mathcal{S} \subseteq \mathcal{T}} f(\mathcal{S}) = \sum_{x_i \in \mathcal{S}} \left[ \alpha \, \mathcal{I}^{\text{in}}(x_i) + \mathcal{I}^{\text{ex}}(x_i | \mathcal{S}) \right], \ \text{s.t. } |\mathcal{S}| = b,$$
$$(4)$$

where the extrinsic importance of $x_i$ is defined as

$$\mathcal{I}^{\text{ex}}(x_i | \mathcal{S}) = \sum_{x_j}^{\mathcal{S} \setminus \{x_i\}} a_{ij} = \sum_{x_j}^{\mathcal{S} \setminus \{x_i\}} g(D(x_i, x_j)). \quad (5)$$

This unified objective provides a sample-wise view of the MWCP, which will be instrumental for designing efficient solver in the following sections.

Despite its conceptual clarity and flexibility, directly optimizing Eq. (4) is computationally prohibitive for large-scale datasets. *First*, MWCP is NP-hard, and exact solvers (Hosseinian et al., 2020) do not scale to realistic dataset sizes. *Second*, constructing a fully connected graph requires computing all pair-wise interactions, leading to $\mathcal{O}(N^2)$ complexity. To address these challenges, we propose an efficient optimization framework consisting of two complementary components: (i) a greedy approximation algorithm for MWCP, which reduces the subset selection complexity, and (ii) a structured graph sparsification strategy, which reduces the cost of computing extrinsic importance by restricting pair-wise interactions.

### 3.4. Greedy Selection with Unified Importance

In this section, we present a greedy approximation algorithm inspired by a key observation on the behavior of the objective when the clique is locally perturbed. That is, although

the MWCP is NP-hard in general, it can still get the optimal solution when the cardinality is set as $b = N - 1$, namely only removing one vertex from $G$. Specifically, removing a vertex $v_i \in C$ reduces the total clique weight by

$$\Delta^-(v_i | G) = w_i + \sum_{v_j \in C \setminus \{v_i\}} a_{ij}, \quad (6)$$

which consists of the vertex weight of $v_i$ and all edge weights incident to $v_i$ within the clique. Hence, among all possible single-vertex deletions, the optimal choice is simply the vertex with the smallest removal cost $\Delta^-(v_i | G)$.

**Unified Importance.** This observation implies that while MWCP is globally intractable, it becomes locally solvable under single-vertex perturbations. Crucially, the quantity in Eq. (6) depends only on the neighborhood of $v_i$ and admits a natural interpretation as a *marginal contribution* to the objective. By symmetry, the same quantity can be interpreted from an additive perspective. Let $\mathcal{S}$ denote the subset of samples corresponding to the clique $C$. Then unified importance $\mathcal{I}(x_i | \mathcal{S})$ the contribution of $x_i$ to the unified objective in Eq. (4):

$$\mathcal{I}(x_i | \mathcal{S}) = \Delta(x_i | \mathcal{S}) = \alpha \, \mathcal{I}^{\text{in}}(x_i) + \mathcal{I}^{\text{ex}}(x_i | \mathcal{S}). \quad (7)$$

This quantity measures the marginal gain of including $x_i$ in the current subset, or equivalently, the marginal loss incurred by removing it, which provides a principled justification for defining the importance score as the marginal gain with respect to our objective. Please note that this is not introduced heuristically, but arises directly from the exact solution of a locally constrained MWCP.

**Greedy Selection Strategy.** Based on the above observation and $\mathcal{I}(x_i | \mathcal{S})$, we propose a greedy approximation that incrementally constructs the subset $\mathcal{S}$ by prioritizing samples with large marginal gains. Starting from an initial set, at each iteration we select the sample that maximizes Eq. (7) with respect to the current subset $\mathcal{S}_t$. Formally, let $\mathcal{S}_t$ denote the subset at iteration $t$. The greedy update rule is given by

$$x^\star \leftarrow \arg \max_{x_i \in \mathcal{T} \setminus \mathcal{S}_t} \mathcal{I}(x_i \mid \mathcal{S}_t), \quad \mathcal{S}_{t+1} \leftarrow \mathcal{S}_t \cup \{x^\star\}, \quad (8)$$

which repeats until $|\mathcal{S}| = b$. This algorithm has linear selection complexity in the subset size and avoids the combinatorial explosion of exact solvers. In practice, we employ an efficient implementation as shown in Algorithm 1. Moreover, when the edge weights satisfy certain regularity conditions, the unified objective exhibits favorable properties that allow the greedy solution to admit a provable $(1 - \frac{1}{e})$ worst-case approximation guarantee. These theoretical analyses are formally established in Sec. 3.6. The practical implementation with graph sparsification technique of this greedy solver is detailed in Algorithm 1. Additionally, for very large datasets, we propose a *Stochastic Selection* strategy,

**Algorithm 1** Overall pipeline with Greedy Selection and Structured Graph Sparsification

---

**Require:** Training dataset $\mathcal{T} = \{x_i\}_{i=1}^{N}$, pruning ratio $p$,
**Ensure:** Pruned subset $\mathcal{S}$.
1: Build a fully connected graph $G = (V, E)$ from $\mathcal{T}$.
2: Compute intrinsic importance $\mathcal{I}^{in}(x_i)$ for all $x_i \in \mathcal{T}$.
3: Perform Structured Graph Sparsification and get neighborhood clusters $\{\mathcal{N}_k\}_{k=1}^{K}$.
4: Compute $\{D(x_i, x_j) \mid \forall x_i, x_j \in \mathcal{N}_k\}$ for all $\mathcal{N}_k$.
5: Initialize $\mathcal{S}_0 \leftarrow \varnothing$, $\mathcal{I}^{ex}(x_i|\mathcal{S}_0) \leftarrow 0$ for all $x_i$.
6: **for** $t = 1$ **to** $(1 - p)N$ **do**
7:     **for all** $x_i \in \mathcal{T} \setminus \mathcal{S}_{t-1}$ **do**
8:         **if** $x^\star \in \mathcal{N}(x_i)$ **then**
9:             Fetch $\mathcal{I}^{in}(x_i)$ and $D(x_i, x^\star)$.
10:             $\mathcal{I}^{ex}(x_i|\mathcal{S}_t) \leftarrow \mathcal{I}^{ex}(x_i|\mathcal{S}_{t-1}) + g(D(x_i, x^\star))$
11:             $\mathcal{I}(x_i|\mathcal{S}_t) \leftarrow \alpha \mathcal{I}^{in}(x_i) + \mathcal{I}^{ex}(x_i|\mathcal{S}_t)$
12:         **end if**
13:     **end for**
14:     Select $x^\star \leftarrow \arg\max_{x_i \in \mathcal{T} \setminus \mathcal{S}} \mathcal{I}(x_i|\mathcal{S}_t)$.
15:     Update $\mathcal{S}_t \leftarrow \mathcal{S}_{t-1} \cup \{x^\star\}$.
16: **end for**
17: **Return** $\mathcal{S} \leftarrow \mathcal{S}_{(1-p)N}$.

---

which computes normalized $\mathcal{I}(x_i|\mathcal{T} \setminus \{x_i\})$ as probability for stochastic sampling, offering a faster alternative. Please refer to Sec. C for details.

### 3.5. Structured Graph Sparsification

The unified objective in Eq. (4) involves all pairwise interactions over the entire dataset. To reduce the computational cost incurred by all edges of $G$, we introduce the Structured Graph Sparsification that reduces the number of edges while also keeping the form of the MWCP objective.

We associate each sample $x_i$ with a neighborhood $\mathcal{N}(x_i) \subset \mathcal{T}$, within which extrinsic interactions are evaluated. Accordingly, the extrinsic importance term is computed as

$$\mathcal{I}^{ex}(x_i \mid \mathcal{S}) = \sum_{x_j \in \mathcal{S} \cap \mathcal{N}(x_i)} g(D(x_i, x_j)), \qquad (9)$$

where $\mathcal{N}(x_i)$ defines the interaction support of $x_i$. We define $\mathcal{N}(x_i)$ through a two-level structure. First, samples are partitioned by class labels, which reduces the scale of candidate interactions. Within each class, samples are further clustered based on their features, and $\mathcal{N}(x_i)$ is defined as the cluster to which $x_i$ belongs. For datasets without any class labels, we can simply divide it solely by clustering. This construction reflects the observation that redundancy is predominantly local, and that samples outside $\mathcal{N}(x_i)$ typically contribute negligible extrinsic influence.

From a graph-theoretic perspective, this procedure induces a sparse weighted graph whose edge set corresponds to the

restricted interaction supports $\{\mathcal{N}(x_i)\}$. Edges absent from this graph can be equivalently interpreted as having zero weight, i.e., $a_{ij} = 0$. Hence, the sparsified graph still remains formally equivalent to a fully connected graph with a large number of zero-weight edges with less actual edges and computation. As a result, all formulations and derivations in previous sections, including the MWCP and the greedy solver, remain valid without modification. Pruning pipeline with this sparsification is shown in the Algorithm 1. Please refer to Sec. G for the detailed complexity analysis.

### 3.6. Theoretical Analysis and Design Guidance

This section provides theoretical justification for the proposed greedy solver. We show that under mild and easily satisfied conditions, the unified pruning objective in Eq. (4) satisfies a submodular property, under which greedy optimization enjoys a provable approximation guarantee..

**Definition 3.3** (Submodularity). A set function $f : 2^{\mathcal{T}} \rightarrow \mathbb{R}$ is said to be *submodular* if it satisfies the diminishing-returns property: for all $\mathcal{A} \subseteq \mathcal{B} \subseteq \mathcal{T}$ and any $x \notin \mathcal{B}$,

$$\Delta(x \mid \mathcal{A}) \geq \Delta(x \mid \mathcal{B}), \qquad (10)$$

where the marginal gain is defined as

$$\Delta(x \mid \mathcal{A}) = f(\mathcal{A} \cup \{x\}) - f(\mathcal{A}). \qquad (11)$$

**Lemma 3.4** (Submodularity of the unified pruning objective). *Suppose $D(\cdot, \cdot)$ is a non-negative distance metric and $g : \mathbb{R}_{\geq 0} \rightarrow \mathbb{R}_{\leq 0}$ is non-positive on $[0, +\infty)$. Then the unified objective $f(\mathcal{S})$ in Eq.(4) is submodular.*

**Proof.** For any $\mathcal{A} \subseteq \mathcal{B}$ and $x \notin \mathcal{B}$, the marginal gains of adding $x$ can be written as:

$$\Delta(x|\mathcal{A}) = \mathcal{I}(x|\mathcal{A}) = \alpha \mathcal{I}^{in}(x) + \sum_{y \in \mathcal{A}} g(D(x, y)),$$
$$\Delta(x|\mathcal{B}) = \mathcal{I}(x|\mathcal{B}) = \alpha \mathcal{I}^{in}(x) + \sum_{y \in \mathcal{B}} g(D(x, y)). \qquad (12)$$

Their difference is

$$\Delta(x|\mathcal{A}) - \Delta(x|\mathcal{B}) = -\sum_{y \in \mathcal{B} \setminus \mathcal{A}} g(D(x, y)). \qquad (13)$$

Since $D(\cdot, \cdot) \geq 0$ and $g(\cdot) \leq 0$ on $[0, +\infty)$, every term $g(D(x, y)) \leq 0$, which implies

$$\Delta(x|\mathcal{A}) - \Delta(x|\mathcal{B}) \geq 0. \qquad (14)$$

Hence $\Delta(x|\mathcal{A}) \geq \Delta(x|\mathcal{B})$ for all $\mathcal{A} \subseteq \mathcal{B}$, establishing the submodularity of $f(\mathcal{S})$. $\square$

This lemma formalizes a mild condition that ensure the submodular structure of the unified objective: a non-negative distance metric and a non-positive mapping.

*Table 1.* Comparison to state-of-the-art methods on CIFAR-10/100 using ResNet-18. The proposed method achieves the best performance. * denotes different prune ratio settings. † denotes our reproduction.

| Dataset | CIFAR-10 | | | CIFAR-100 | | |
|---|---|---|---|---|---|---|
| **Pruning Ratio** | 30% | 50% | 70% | 30% | 50% | 70% |
| Random | 94.6 | 93.3 | 90.2 | 73.8 | 72.1 | 69.7 |
| RS2 (Okanovic et al., 2024) | - | 95.2 | 94.3 | 78.3 | 77.6 | 76.1 |
| Herding(Chen et al., 2012) | 92.2 | 88.0 | 80.1 | 73.1 | 71.8 | 69.6 |
| Influence(Koh & Liang, 2017) | 93.1 | 91.3 | 88.3 | 74.4 | 72.0 | 68.9 |
| K-Center(Sener & Savarese, 2018) | 94.7 | 93.9 | 90.9 | 74.1 | 72.2 | 70.2 |
| SVP(Coleman et al., 2019) | 95.0 | 94.5 | 90.3 | 74.2 | 72.3 | 69.8 |
| Craig(Mirzasoleiman et al., 2020) | 94.8 | 93.3 | 88.4 | 74.4 | 71.9 | 69.7 |
| SB-12hours†(Jiang et al., 2019) | 95.5 | 95.1 | 93.2 | - | - | - |
| GraNd(Paul et al., 2021) | 95.3 | 94.6 | 91.2 | 74.6 | 71.4 | 68.8 |
| Glister(Killamsetty et al., 2021b) | 95.2 | 94.0 | 90.9 | 74.6 | 73.2 | 70.4 |
| $\epsilon$-greedy(Raju et al., 2021) | 95.2 | 94.9 | 94.1 | 76.4 | 74.8 | - |
| UCB(Raju et al., 2021) | 95.3 | 94.7 | 93.9 | 77.3 | 75.3 | - |
| Forgetting(Toneva et al., 2018a) | 94.7 | 94.1 | 91.7 | 75.3 | 73.1 | 69.9 |
| EL2N(Paul et al., 2021) | 95.3 | 95.1 | 91.9 | 77.2 | 72.1 | - |
| Training Loss(Kawaguchi & Lu, 2020) | - | - | 94.6 | - | - | 72.6 |
| AUM(Pleiss et al., 2020) | 95.1 | 95.3 | 91.4 | 76.9 | 67.4 | 30.6 |
| Moderate(Xia et al., 2022) | 93.7 | 92.6 | 90.6 | 74.3 | 68.3 | 57.8 |
| (Katharopoulos & Fleuret, 2018) | - | - | 94.4 | - | - | 73.2 |
| *Dataset Pruning*(Yang et al., 2022) | 94.9 | 93.8 | 90.8 | 77.2 | 73.1 | - |
| CCS(Zheng et al., 2023) | 95.4 | 95.0 | 93.0 | 77.1 | 74.5 | 68.9 |
| MoSo†(Tan et al., 2024) | - | - | - | 76.7 | 72.3 | 65.8 |
| InfoBatch*(Qin et al., 2023) | 95.6 | 95.1 | 94.7 | 78.2 | 78.1 | 76.5 |
| DivBS†(Hong et al., 2024) | 95.4 | 95.2 | 95.1 | 78.5 | 78.2 | 77.2 |
| **Ours** | **95.9** | **95.4** | **95.2** | **78.9** | **78.6** | **77.6** |
| | ↑0.3 | ↓0.2 | ↓0.4 | ↑0.7 | ↑0.4 | ↓0.6 |
| Full Data | 95.6±0.1 | | | 78.2±0.1 | | |

*Table 2.* Performance Comparison on ImageNet-1k. ResNet-50 and Swin-T are trained from scratch for 90 and 300 epochs, respectively. The results demonstrate the effectiveness of the proposed method on both CNNs and Transformers.

| | **Pruning Ratio** | 30% | 50% | 70% |
|---|---|---|---|---|
| ResNet-50 | Random | $72.2_{\downarrow4.2}$ | $69.1_{\downarrow7.3}$ | $65.9_{\downarrow10.5}$ |
| | Herding(Chen et al., 2012) | $73.5_{\downarrow2.9}$ | $69.3_{\downarrow7.1}$ | $65.1_{\downarrow11.3}$ |
| | Forgetting(Toneva et al., 2018a) | $74.8_{\downarrow1.6}$ | $72.0_{\downarrow4.4}$ | $67.8_{\downarrow8.6}$ |
| | EL2N(Paul et al., 2021) | $74.3_{\downarrow2.1}$ | $68.5_{\downarrow7.9}$ | $54.8_{\downarrow21.6}$ |
| | $\mathcal{D}^2$-pruning†(Maharana et al., 2024) | $75.0_{\downarrow1.4}$ | $73.4_{\downarrow3.0}$ | $68.2_{\downarrow8.2}$ |
| | InfoMax†(Tan et al., 2025) | $75.2_{\downarrow1.2}$ | $73.7_{\downarrow2.7}$ | $68.6_{\downarrow7.8}$ |
| | Moderate(Xia et al., 2022) | $75.2_{\downarrow1.2}$ | $72.2_{\downarrow4.2}$ | $67.7_{\downarrow8.7}$ |
| | MoSo†(Tan et al., 2024) | $76.5_{\uparrow0.1}$ | $73.5_{\downarrow2.9}$ | $70.0_{\downarrow6.4}$ |
| | InfoBatch†(Qin et al., 2023) | $76.5_{\uparrow0.1}$ | $75.8_{\downarrow0.6}$ | $74.9_{\downarrow1.5}$ |
| | **Ours** | $\mathbf{77.0_{\uparrow0.6}}$ $\pm0.1$ | $\mathbf{76.5_{\uparrow0.1}}$ $\pm0.1$ | $\mathbf{75.3_{\downarrow1.1}}$ $\pm0.2$ |
| | Full Data | 76.4±0.2 | | |

| | **Pruning Ratio** | 30% | 40% | 50% |
|---|---|---|---|---|
| Swin-T | Random | $77.2_{\downarrow2.4}$ | $75.9_{\downarrow3.7}$ | $74.5_{\downarrow5.1}$ |
| | Forgetting(Toneva et al., 2018a) | $78.3_{\downarrow1.3}$ | $77.6_{\downarrow2.0}$ | $74.3_{\downarrow5.3}$ |
| | EL2N(Paul et al., 2021) | $78.2_{\downarrow1.4}$ | $75.9_{\downarrow3.7}$ | $71.1_{\downarrow8.5}$ |
| | SVP(Coleman et al., 2019) | $76.6_{\downarrow3.0}$ | $74.9_{\downarrow4.7}$ | $72.7_{\downarrow6.9}$ |
| | Moderate(Xia et al., 2022) | $77.1_{\downarrow2.5}$ | $75.9_{\downarrow3.7}$ | $75.0_{\downarrow4.6}$ |
| | InfoBatch†(Qin et al., 2023) | $78.6_{\downarrow1.0}$ | $78.2_{\downarrow1.4}$ | $77.5_{\downarrow2.1}$ |
| | Dyn-Unc(He et al., 2024) | $79.1_{\downarrow0.5}$ | $78.5_{\downarrow1.1}$ | $77.6_{\downarrow2.0}$ |
| | PFB(Wu et al., 2025) | $79.6_{\uparrow0.0}$ | $79.2_{\downarrow0.4}$ | $78.2_{\downarrow1.4}$ |
| | **Ours** | $\mathbf{79.7_{\uparrow0.1}}$ $\pm0.1$ | $\mathbf{79.4_{\downarrow0.2}}$ $\pm0.1$ | $\mathbf{78.9_{\downarrow0.7}}$ $\pm0.1$ |
| | Full Data | 79.6±0.1 | | |

**Design Flexibility and Example instantiation.** The submodularity holds for a large family of $\mathcal{I}^{in}(\cdot)$ and $D(\cdot,\cdot)$, as long as the induced pairwise interaction is non-positive. This flexibility allows the unified importance to accommodate diverse criteria such as loss, uncertainty, or forgetting scores, while preserving theoretical guarantees. We provide an instantiation satisfying Lemma 3.4 as an example:

$$\mathcal{I}^{in}(x_i) = H(\mathbf{p}_i) = -\sum_{c=1}^{K} p_i^c \log p_i^c,$$
$$\mathcal{I}^{ex}(x_i|\mathcal{S}) = \sum_{x_j \in \mathcal{S}} \left[ \phi\big(D^{cos}(x_i, x_j)\big) - 1 \right], \quad (15)$$

where $H(\mathbf{p}_i)$ is the entropy of the predicted probability distribution $\mathbf{p}_i \in \mathbb{R}^{1 \times K}$ of sample $x_i$, $K$ is the number of classes; and $\phi(\cdot)$ denotes the sigmoid function. Intuitively, with these settings, maximizing $f(\mathcal{S})$ leads to selecting a subset with large intrinsic information content and strong sample diversity. *Please note that this is only an effective example instantiation. It does not depend on any particular fixed choice.* See Sec. B and Sec. D.3 for more instantiations.

**Greedy approximation guarantee.** We now recall the classical result of Nemhauser et al. (Nemhauser et al., 1978), which characterizes the performance of greedy optimization on monotone submodular functions.

**Theorem 3.5** (Nemhauser et al. (Nemhauser et al., 1978))**.** *Let $f$ be a monotone submodular set function, namely for*

*any two sets $\mathcal{A}, \mathcal{B} \subseteq \mathcal{T}$ satisfying $\mathcal{A} \subseteq \mathcal{B}$, the following holds:*

$$f(\mathcal{A}) \leq f(\mathcal{B}). \quad (16)$$

*Under the constraint $|\mathcal{S}| = b$, the greedy solution $\mathcal{S}$ satisfies*

$$f(\mathcal{S}) \geq \left(1 - \frac{1}{e}\right) f(\mathcal{S}^*), \quad (17)$$

*where $\mathcal{S}^*$ is the optimal subset of size $b$.*

In our case, $f(\mathcal{S})$ is submodular by Lemma 3.4. Monotonicity can be ensured by adding a constant modular term that upper-bounds the magnitude of the pairwise penalty. Such a transformation does not affect the greedy selection order and is omitted in practice. Details are provided in the Sec. E.

**Design Guidance.** Although the original MWCP formulation is NP-hard, the unified objective admits a submodular structure under the mild conditions:

1. *a non-negative distance metric $D(\cdot, \cdot)$ and a mapping $g(\cdot)$ that is non-positive on $[0, +\infty)$.*

2. *non-negative marginal gains $\mathcal{I}(x|\mathcal{A})$ for any $x$ and $\mathcal{A}$.*

Consequently, the greedy solver used in our algorithm enjoys a constant-factor approximation guarantee. From a practical perspective, this analysis provides clear guidance for importance design: any choice of intrinsic and extrinsic scores that induces non-positive pairwise interactions

*Table 3.* Time cost to reach the same Acc performance on ImageNet-1k using ResNet-50. Our method achieves lossless training acceleration and saves the overall cost by 46.5%. Our results are obtained under a 50% pruning ratio. EL2N-50 and our method need a preprocess phase. EL2N-50 need to train a model from scratch for 50 epochs to collect the L2 prediction error for each sample. Our method need to extract features for the graph construction and the computation of extrinsic importance.

| Method | Year | Pruning Freq. | Acc(%) | Training(h) | Pruning Overhead(h) preprocess | pruning | Total Time | Reduction |
|---|---|---|---|---|---|---|---|---|
| Full Data | - | - | 76.4 | 13.9 | - | - | 13.9 | - |
| EL2N-50[†](Paul et al., 2021) | 2018 | Single-shot | 71.5$_{\downarrow4.9}$ | 10.1 | 7.72 | 0.03 | 17.9 | 28.8%↑ |
| UCB[†](Raju et al., 2021) | 2021 | Epoch | 75.8$_{\downarrow0.6}$ | 10.1 | - | 0.08 | 10.2 | 26.6%↓ |
| InfoBatch[†](Qin et al., 2023) | 2023 | Epoch | **76.6**$_{\uparrow0.2}$ | 10.1 | - | 0.07 | 10.2 | 26.6%↓ |
| DivBS[†](Hong et al., 2024) | 2024 | Batch | 76.4$_{\uparrow0.0}$ | 11.2 | - | 0.72 | 11.9 | 14.4%↓ |
| PFB[†](Wu et al., 2025) | 2025 | Batch | 76.4$_{\uparrow0.0}$ | 8.6 | - | **0.06** | 8.7 | 37.4%↓ |
| **Ours** | - | Epoch | **76.5**$_{\uparrow0.1}$ | **7.3** | 0.55 | 0.07 | **7.9** | **43.2%↓** |

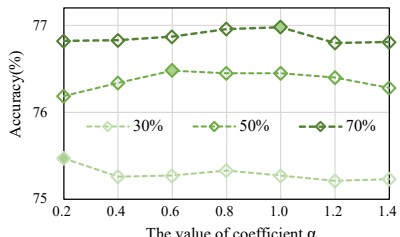

*Figure 1.* Ablation on coefficient $\alpha$. The best results are highlighted using solid markers.

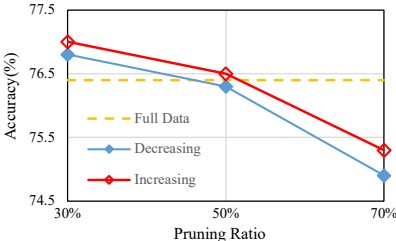

*Figure 2.* Ablation on the monotonicity of extrinsic importance.

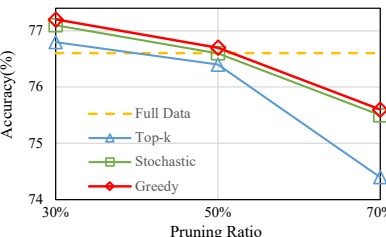

*Figure 3.* Ablation on different solvers for the MWCP objective.

and non-negative marginal gains leads to a theoretically grounded and scalable pruning algorithm.

# 4. Experiments

## 4.1. Experimental Setup

**Datasets.** Following the most common setting of the latest works(He et al., 2024; Hong et al., 2024; Qin et al., 2023), we conduct experiments to evaluate the performance of UGIES on CIFAR-10(Krizhevsky et al., 2009), CIFAR-100(Krizhevsky et al., 2009), and ImageNet-1k(Deng et al., 2009) for image classification. CIFAR-10/100 datasets consist of 32×32 sized images that are divided into 10 and 100 classes, respectively. Each of them contains 50,000 images for training and 10,000 for testing. ImageNet-1k is a large dataset containing 1,281,167 training images and 50,000 validation images that are categorized into 1,000 classes.

**Implementation details.** Our method follows an epoch-wise dynamic pruning paradigm. We employ ResNet-18(He et al., 2016) as a backbone on CIFAR-10/100 trained for 200 epochs. The batch size is set as 128. On ImageNet-1k, we evaluate the performance of UGIES using ResNet-50 and Swin-T(Liu et al., 2021) to evaluate the generalization across different network architectures. The batch size of both ResNet-50 and Swin-T is set as 1024. Other detailed settings can be found in the Sec. H.

**Settings of $\mathcal{I}^{in}$ and $\mathcal{I}^{ex}$.** As discussed in Sec. 3, our framework supports many intrinsic and extrinsic importance functions as long as they satisfy the mild conditions in Lemma 3.4. Although Eq. 15 provides an effective instance, different tasks or pruning ratios may favor different forms of $\mathcal{I}^{in}$ and $\mathcal{I}^{ex}$, motivating the flexibility choice.

## 4.2. Comparisons with SOTA Methods

We compare UGIES with state-of-the-art dataset pruning methods on CIFAR-10/100 and ImageNet-1k, and report Top-1 accuracy of models trained on the selected subset $\mathcal{S}$. Following standard practice, we use the pruning ratio $p = \frac{|\mathcal{T}\setminus\mathcal{S}|}{|\mathcal{T}|}$ to measure data reduction. For batch-wise pruning methods (Hong et al., 2024; Jiang et al., 2019; Wu et al., 2025), the pruning ratio is defined within each mini-batch as $p = \frac{N_S}{N_B}$, where $N_B$ is the batch size and $N_S$ the number of retained samples. Please also note that InfoBatch (Qin et al., 2023) applies pruning only to a low-loss subset, making its effective pruning ratio roughly half of the reported value. We therefore denote its original results using '*' and our reproduction using '†', where 95% of samples are treated as pruning candidates for consistency. Random pruning is included as a baseline, and all reproduced results under our training setup are marked with '†'.

**Performance Comparisons.** We report the Top-1 accuracy (Acc) results of ResNet-18 on CIFAR-10/100 and

*Table 4.* Ablation of the intrinsic and extrinsic term. Combining both of them yields better results across all pruning ratios.

| $\mathcal{I}(x_i\|\mathcal{S})$ | | Pruning Ratio | | |
|---|---|---|---|---|
| $\mathcal{I}^{in}(x_i)$ | $\mathcal{I}^{ex}(x_i\|\mathcal{S})$ | 30% | 50% | 70% |
| ✓ | | 76.6 | 76.0 | 74.4 |
| | ✓ | 76.3 | 76.1 | 75.3 |
| ✓ | ✓ | **77.0** | **76.5** | **75.4** |
| Full Data | | | $76.4_{\pm0.1}$ | |

*Table 5.* Ablation on different intrinsic importance. The details are shown in Sec.B.

| $\mathcal{I}^{in}(x_i)$ | 30% | 50% | 70% |
|---|---|---|---|
| Entropy | **77.0** | **76.5** | 75.2 |
| Loss | 76.8 | **76.5** | 75.5 |
| Gradient Norm | 76.8 | 76.1 | 74.9 |
| Loss Variation | 76.6 | 76.3 | 74.9 |
| Entropy Variation | 76.8 | 76.2 | 74.7 |
| Loss × Entropy | 76.8 | 76.2 | 75.2 |
| Full Data | | $76.4_{\pm0.1}$ | |

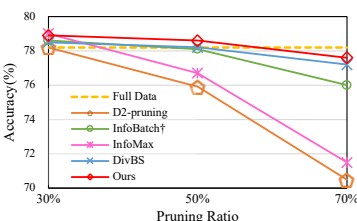

*Figure 4.* Comparison across pruning method families on CIFAR-100. The proposed method excels not only over intrinsic and extrinsic ones, but also over hybrid methods.

ResNet-50 on ImageNet-1k are reported in Table 1 and Table 2, respectively. On CIFAR-10/100, the proposed method clearly outperforms existing methods. Our method achieves better-than-lossless training under 30% pruning ratio, which improves the Acc by 0.3%, 0.7%, and 0.6% over full-data training on CIFAR-10, CIFAR-100, and ImageNet-1k, respectively. This improvement could be attributed to the fact that $\mathcal{S}$ is more informative and less redundant than $\mathcal{T}$. A detailed interpretation can be found in our appendix. These results strongly validate the superiority of the unified objective and the effectiveness of the corresponding greedy solver. Moreover, UGIES achieves lossless training on only 50% data on ImageNet-1k with millions of samples, which demonstrates the generalization ability of our method on large modern datasets. UGIES also generalizes well on Swin-T, achieving lossless acceleration by reducing 30% of training samples.

**Efficiency Comparisons.** Since our approach aims at achieving acceleration, we compare the time cost throughout the training process of several state-of-the-art methods in Table 3. Results are collected on a 4-RTX 4090 GPU server. Automatic Mixed Precision (Micikevicius et al., 2018) is adopted during training for all methods. 'Training' denotes the wall clock time spent on network training procedure, including data loading, forward, and backward. 'Overheads' is the additional time brought in by dataset pruning methods. As our method also need to build graph and compute distance within clusters before training, we report this time cost as 'preprocess'. The results show that our method reduces training time by more than 45% while maintaining the same accuracy as full-set training. Moreover, its clear speed advantage over other state-of-the-art methods highlights the effectiveness of our unified importance evaluation scheme and the efficiency of the proposed greedy solver. We attribute this to the principled objective and the theoretically grounded solver.

### 4.3. Ablations and Analysis

To examine how different importance designs affect pruning behavior, we conduct a series of ablation studies using mul-

tiple intrinsic and extrinsic formulations. These experiments evaluate the design principles enabled by our unified objective under varying pruning budgets. Unless otherwise stated, all results are reported on ImageNet-1k with ResNet-50.

**Ablation on intrinsic and extrinsic terms.** We first examine how the two components of our unified importance affect performance in Table 4. The intrinsic term dominates at low pruning ratios, while the extrinsic term becomes increasingly beneficial under aggressive pruning. When combined in UGIES, the two terms complement each other and surpass either one alone across all pruning regimes, highlighting the value of jointly modeling intrinsic and extrinsic terms.

**Different choices of intrinsic importance.** We evaluate various intrinsic importance within UGIES in Tab. 5 and find that the framework achieves decent performance across all variants, showing robustness to this design choice. Loss and entropy are generally the most stable, yet they differ in their strengths: entropy tends to work better at low pruning ratios, whereas loss is more effective under aggressive pruning. This complementary behavior underscores the value of UGIES as a flexible framework that allows selecting the intrinsic metric best suited to the pruning regime.

**Effect of the intrinsic–extrinsic balance.** Fig. 1 illustrates how the coefficient $\alpha$ influences the balance between intrinsic rewards and extrinsic diversity. No single fixed value of $\alpha$ performs best across all pruning ratios: intrinsic-heavy settings work well at low pruning levels, whereas diversity-oriented configurations clearly dominate under aggressive pruning. This ratio-dependent shift indicates that the optimal balance is not universal but varies with the pruning budget. Such behavior highlights the necessity of a flexible framework—like UGIES—that can adapt the intrinsic-extrinsic tradeoff rather than relying on a fixed metric.

**Ablation on Extrinsic Importance.** As discussed in Sec. 3.6, once the mapping $g$ and distance $D$ satisfy the Lemma 3.4, our greedy solver enjoys the approximation guarantee. Among the many valid choices of $(g, D)$, the most crucial behavioral factor is the monotonicity of the extrinsic term with respect to $D$, which determines whether

the model favors diversity (monotone-increasing) or similarity (monotone-decreasing). Fig. 2 shows that promoting diversity yields higher accuracy, and the advantage becomes more pronounced at higher pruning ratios. This suggests that, diversity-oriented formulations are generally more beneficial. We also explore various $\mathcal{I}^{ex}(x_i|\mathcal{S})$ in Sec. D.3

**Ablation on the Greedy Solver.** The theoretically guaranteed greedy solver for the MWCP is one of the key contributions of our approach. To verify its effectiveness, we compare it with the commonly used *top-k* selection strategy. The results in Fig. 3 show a clear performance gap between the two methods: our greedy solver consistently achieves superior results against Top-K across various pruning ratios, which highlights the advantage of the theoretically grounded optimization algorithm. In addition, the results of *stochastic selection* variant suggest a decent trade-off between efficiency and accuracy. Since this strategy incurs less overheads, it could be a practical choice for very large datasets. Please refer to Sec. C for details.

**Comparison with Different Pruning Method Families.** We further compare UGIES with representative intrinsic (InfoBatch), extrinsic (DivBS), and graph-based methods ($D^2$-Pruning, InfoMax) across pruning ratios (Fig. 4). Intrinsic and extrinsic baselines each perform well only in their favorable regimes—InfoBatch at low pruning ratios and DivBS at high ratios—while both degrade sharply outside these regions. Graph-based methods improve stability by incorporating relational cues, but their heuristic formulations cause inconsistent behavior as the pruning ratio increases. Our UGIES achieves the most stable performance across all settings. Because its importance scores arise directly from a principled intrinsic–extrinsic objective, rather than a fixed metric or heuristic message passing, the method naturally adjusts to different pruning budgets: intrinsic signals dominate when pruning is mild, while relational diversity becomes decisive under aggressive compression. This unified objective explains why UGIES remains closest to or even higher than the full-data accuracy line under all ratios.

## 5. Limitation and Future Work

Despite its effectiveness, UGIES still has several limitations. First, modeling pairwise sample interactions introduces additional computational and memory overhead, especially for extremely large-scale datasets. Although our structured graph sparsification substantially reduces this cost, further improving scalability through more efficient approximation or implicit interaction modeling remains an important future direction. Second, our approximation guarantee relies on mild conditions on the extrinsic importance metrics. These conditions cover a broad family of useful designs and provide practical guidance, but may require proper mappings for arbitrary metrics. Finally, our greedy solver provides

an efficient approximation rather than an exact solution to the underlying combinatorial objective. Exploring tighter solvers or adaptive selection strategies may further improve pruning quality in more challenging settings.

## 6. Conclusion

We introduced UGIES, a unified graph-based framework for dataset pruning that jointly models intrinsic and extrinsic importance. By reformulating the pruning objective as the MWCP, we derived a principled marginal gain importance and an efficient greedy solver. Under mild and interpretable conditions, the objective becomes submodular and monotone, yielding a pruning procedure with formal performance guarantees. Extensive experiments demonstrate that UGIES consistently improves accuracy–efficiency trade-offs and provides a flexible foundation for selecting or designing importance metrics across models, datasets, and pruning ratios according to their specific demands.

## Acknowledgments

This work was supported by Ant Group Research Intern Program. This work made use of computational resources provided by both the Ant Group and the HPC Platform of Huazhong University of Science and Technology. We thank Wang Rui and Dr. Wang Jin for their helpful discussions and suggestions.

## Impact Statement

This work aims to improve the efficiency of training modern deep learning models by reducing the size of training datasets through principled dataset pruning. By identifying and retaining the most informative samples, the proposed framework can substantially reduce computational cost, training time, and energy consumption. As large-scale models continue to grow in size and data requirements, such improvements contribute to lowering the environmental footprint of machine learning and improving the accessibility of training resource-intensive models to institutions with limited computational resources. We believe that by offering a transparent, theoretically grounded formulation and explicit design conditions, our framework can facilitate more informed and responsible use of dataset pruning in practice.

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

# Appendix

This appendix provides additional details and results to complement the main paper. First, interpretations on the accuracy improvement of our method and other existing methods (such as InfoBatch) are presented (Sec. A). Next, we introduce the full definitions of the intrinsic importance functions used in our unified framework (Sec. B). The details of the proposed statistic selection in the Tab. 3 of the main text are provided (Sec. C). We then present extended experimental evaluations, including results on semantic segmentation, object detection, and several ablation studies on the design choices of extrinsic importance and feature sources (Sec. D). Next, we provide a detailed discussion on the simple method to maintaining the monotonicity of the objective function (Sec. E). We further give the complete proof of the $(1 - 1/e)$ greedy approximation guarantee used in our framework (Sec. F). The detailed complexity of the greedy selection is discussed (Sec. G). Finally, we describe additional implementation details for reproduction of our experiments (Sec. H).

## A. Interpretation on Performance Improvement after Pruning

Interestingly, we observe that pruning a moderate portion of the training data (typically around 30%–50%) with our proposed method sometimes leads to performance that matches or even surpasses training on the full dataset. This phenomenon can be naturally explained through the lens of our unified pruning objective.

Specifically, our objective aims to maximize the overall unified importance, which jointly accounts for a sample's intrinsic learning utility and its extrinsic redundancy with respect to other samples. The proposed greedy selection strategy prioritizes samples with large marginal gains and progressively removes samples whose effective contribution is limited due to low intrinsic informativeness and strong redundancy. Such samples provide little new information beyond what has already been captured by the current training model and may even interfere with optimization when their losses and gradients are averaged within mini-batches. As a result, the retained subset exhibits higher information density and reduced redundancy, enabling the model to focus on more informative and representative training signals while preserving generalization ability. Furthermore, some pruned samples may implicitly contain noise or spurious correlations, whose removal further stabilizes training and improves performance. Consequently, when the pruning ratio remains within a moderate range, dataset pruning can lead to lossless or even improved performance.

This behavior has also been observed in prior works such as InfoBatch (Qin et al., 2023), indicating that the phenomenon is not unique to our method but rather intrinsic to importance-based pruning strategies. Nevertheless, similar to existing approaches, our method does not achieve lossless acceleration under overly aggressive pruning. When the pruning ratio becomes too large, samples with high intrinsic importance are inevitably discarded, leading to insufficient coverage of the data distribution and a subsequent drop in performance.

## B. Different Intrinsic Importance

This section provides the detailed definitions of the intrinsic importance functions used in Tab. 5 of the main text.

**Entropy.** We use prediction entropy $H(\mathbf{p}_i)$ as a standard uncertainty measure:

$$\mathcal{I}^{in}_{entropy}(x_i) = H(\mathbf{p}_i) = -\sum_{c=1}^{K} p_i^c \log p_i^c, \tag{18}$$

where $K$ is the number of classes and $\mathbf{p}_i$ is the final prediction of $x_i$.

**Loss.** Loss reflects the learning difficulty of each sample. Since most experiments are conducted on image classification, we use cross-entropy loss:

$$\mathcal{I}^{in}_{loss\_cls}(x_i) = \text{CELoss}(\mathbf{p}_i, \mathbf{y}_i), \tag{19}$$

where $\mathbf{y}_i$ is the ground truth label. For other tasks, the same definition can be instantiated using the corresponding objective (e.g., MSE or KL divergence).

**Gradient Norm.** We also explore the gradient norm of the final layer as a representative of gradient-based criteria:

$$\mathcal{I}^{in}_{grad}(x_i) = \|\text{Grad}(\mathbf{W}_{final})\|_2, \tag{20}$$

where $\text{Grad}(\cdot)$ denotes the gradient of the referenced tensor, and $\mathbf{W}_{final}$ denotes the learnable weights of the final layer.

**Entropy and Loss Variation.** We further consider the epoch-wise variations:

$$
\begin{aligned}
\mathcal{I}_{va\_grad}^{in}(x_i) &= \left| \mathcal{I}_{grad}^{in}(x_i)^{t-1} - \mathcal{I}_{grad}^{in}(x_i)^t \right|, \\
\mathcal{I}_{va\_loss}^{in}(x_i) &= \left| \mathcal{I}_{loss}^{in}(x_i)^{t-1} - \mathcal{I}_{loss}^{in}(x_i)^t \right|,
\end{aligned}
\tag{21}
$$

where $t$ and $t-1$ denote consecutive epochs. These quantities capture relative learning progress and help prioritize still-evolving or recently forgotten samples.

**Loss $\times$ Entropy.** This composite score highlights samples that are simultaneously hard and uncertain:

$$
\mathcal{I}_{loss \times entropy}^{in}(x_i) = \mathcal{I}_{loss}^{in}(x_i) \cdot \mathcal{I}_{grad}^{in}(x_i).
\tag{22}
$$

## C. Faster Pruning Implementation: Stochastic Selection via Importance Sampling.

In addition to the deterministic greedy rule, the unified importance scores $\mathcal{I}(x_i|\mathcal{T}\backslash\{x_i\})$, which is equivalent to the marginal loss $\Delta^-(v_i|G)$, can be used to construct a stochastic sampling distribution.

$$
\mathcal{I}(x_i|\mathcal{T}\backslash\{x_i\}) = \Delta^-(v_i|G)
\tag{23}
$$

Specifically, we convert the scores into a probability measure over the dataset:

$$
\pi(x_i) = \frac{\exp\big(\mathcal{I}(x_i)\big)}{\sum_{x_j \in \mathcal{T}} \exp\big(\mathcal{I}(x_j)\big)}
\tag{24}
$$

A pruned subset $\mathcal{S}$ is then obtained by sampling $(1-p)N$ elements *without replacement* according to $\pi(\cdot)$. This stochastic variant eliminates the iterative recomputation in greedy selection and reduces the selection complexity to $O(N)$ after importance computation, making it suitable for extremely large-scale settings.

## D. More Experimental Results

Additional experiments are provided to further illustrate the behavior and generalization of our framework.

### D.1. Performance on Semantic Segmentation

To assess performance on dense prediction tasks, we evaluate our method on the PASCAL VOC 2012 `trainaug` set (Everingham et al., 2010) following the DivBS configuration. Since DivBS performs batch-wise pruning, we adapt our UGIES accordingly and denote this variant as **Ours-BS**. We similarly adapt InfoBatch for fairness (denoted as **InfoBatch-BS**). We evaluate the performance under pruning ratios of 70%, 80%, and 90%, and report final mIoU.

As shown in Tab. 6, Ours-BS consistently achieves the best performance across all pruning rates, demonstrating the strong generalization capability of our unified importance scoring on segmentation tasks.

*Table 6.* Comparison on PASCAL VOC 2012 using ResNet-50 UperNet. † denotes our reproduction.

| Methods | 70% | 80% | 90% |
|---|---|---|---|
| Moderate(Xia et al., 2022) | 68.34 | 66.83 | 63.27 |
| CCS (Silverman, 2018) | 68.47 | 67.22 | 63.98 |
| DivBS (Hong et al., 2024) | 69.85 | 68.13 | 65.45 |
| InfoBatch-BS† (Qin et al., 2023) | 69.80 | 67.97 | 64.66 |
| Ours-BS | 70.02 | 68.48 | 65.79 |
| Full Data | | 70.80 | |

### D.2. Performance on Object Detection

We further evaluate generalization on object detection using SSD-ResNet50 trained on COCO2017 (Lin et al., 2014). We compare with InfoBatch (Qin et al., 2023) and $\mathcal{D}^2$-pruning (Maharana et al., 2024). Tab. 7 shows that our method attains the highest AP across all metrics, indicating strong adaptability to various vision tasks.

*Table 7.* Object detection pruning on COCO2017 using SSD ResNet-50. † denotes our reproduction. The proposed method achieves a large performance improvement over existing methods.

| Method | Pruning Ratio | AP | $AP_{50}$ | $AP_{75}$ | $AP_S$ | $AP_M$ | $AP_L$ |
|---|---|---|---|---|---|---|---|
| Full Data | 0% | 25.2 | 42.7 | 25.8 | 7.3 | 27.1 | 40.8 |
| Random[†] | 25% | 23.5 | 40.4 | 23.8 | 6.1 | 24.9 | 38.9 |
| InfoBatch[†] (Qin et al., 2023) | 25% | 25.0 | 42.3 | 26.0 | 7.6 | 26.9 | 40.7 |
| $\mathcal{D}^2$-pruning[†] (Maharana et al., 2024) | 25% | 24.8 | 41.7 | 25.3 | 7.0 | 26.1 | 41.0 |
| **Ours** | 25% | **25.5** | **42.7** | **26.4** | **7.8** | **27.8** | **41.3** |

## D.3. Ablation on Extrinsic Importance

As discussed in Sec. 3.5, our mild conditions allow a broad family of valid mapping functions $g$ and distance metrics $D$. We therefore provide some choices of mapping functions:

$$
\begin{aligned}
g_1(d) &= \phi(d) - 1, \\
g_2(d) &= -\frac{1}{d + \epsilon}, \\
g_3(d) &= -e^{-d}, \\
g_4(d) &= -\frac{1}{1 + \log(1 + d)}.
\end{aligned}
\tag{25}
$$

We also compare multiple distance metrics (Cosine, L2, L1, L∞). Tab. 8 shows that our method is generally not sensitive to the specific choice of $g$, while cosine distance yields the strongest performance among all $D$. This aligns with the common observation that cosine similarity performs well in high-dimensional feature spaces. Chebyshev distance performs the weakest, likely due to sensitivity to outliers. We also modify other powerful extrinsic pruning method as a option of the extrinsic term in our framework as well. We implement and adapt the gradient similarity of DivBS (Hong et al., 2024) and the probability density of PFB (Wu et al., 2025) to explore metrics beyond a simple distance metric based on features. The results demonstrate that the modified DivBS (without the greedy search) outperforms the others on both low and high prune ratios. However, as the gradient calculation cost extra time, we recommend using cosine distance for the better trade-off between accuracy and efficiency. Please note that we adjusted the $\alpha$ for each setting and report the optimal performance.

*Table 8.* Ablations on different choices of $g(\cdot)$ and $D(\cdot, \cdot)$ under 30% pruning on ImageNet. We also adapt the gradient similarity of DivBS and probability density based on interactions to cluster centers of PFB in our framework as different choice of extrinsic importance.

| Mapping function $g$ | Metric $D$ | 30% | 70% |
|---|---|---|---|
| $g_1(d) = \phi(d) - 1$ | | 77.0 ↑ 0.6 | 75.3 ↓ 1.1 |
| $g_2(d) = -\frac{1}{d+\epsilon}$ | Cosine Distance | 76.9 ↑ 0.5 | 74.9 ↓ 1.5 |
| $g_3(d) = -e^{-d}$ | | 77.0 ↑ 0.6 | 75.2 ↓ 1.2 |
| $g_4(d) = -\frac{1}{1+\log(1+d)}$ | | 76.9 ↑ 0.5 | 75.3 ↓ 1.1 |
| | Gradient Similarity (DivBS w/o greedy) | 77.0 ↑ 0.6 | 75.5 ↓ 0.9 |
| $g_1(d) = \phi(d) - 1$ | Probability Density (PFB w/o random term) | 76.9 ↑ 0.5 | 75.3 ↓ 1.1 |
| | L2(Euclidean) | 76.8 ↑ 0.4 | 75.2 ↓ 1.2 |
| | L1(Manhattan) | 76.8 ↑ 0.4 | 75.1 ↓ 1.3 |
| | L∞(Chebyshev) | 76.6 ↑ 0.2 | 74.6 ↓ 1.8 |
| Full Data | | $76.4_{\pm 0.1}$ | |

## D.4. Ablation on Feature Source for Distance Computation and Graph Construction

Since our extrinsic importance computation depends on sample embeddings, we study the influence of different models to extract features: (i) ImageNet-pretrained ResNet-34 (TorchVision), (ii) ImageNet-pretrained ResNet-50 (TorchVision), and

(iii) the current training model. Tab. 9 shows that all three variants achieve similar accuracy, while the dynamic features (iii) incur better accuracy but also higher overhead due to per-epoch feature extraction and graph rebuilding. Thus, for efficiency, we simply use the setting (ii) for all of our ResNet-50 experiments. Switching to ResNet-34 slightly decreases performance but remains competitive, indicating robustness to the embedding extractor.

*Table 9.* Ablations on feature sources under 30% pruning on ImageNet.

| Feature Source | Architecture | Top-1 Acc. |
| --- | --- | --- |
| TorchVision Pretrained | ResNet-34 | $76.82_{\pm 0.23}$ |
| TorchVision Pretrained | ResNet-50 | $76.96_{\pm 0.11}$ |
| Ours During Training | ResNet-50 | $77.02_{\pm 0.13}$ |

## E. Maintain Monotonicity of the Objective Function

As discussed in Sec. 3.5, we provide a simple approach to ensure $f(\mathcal{S})$ remains monotone. Adding a constant to intrinsic importance, $\mathcal{I}^{in}_{revised}(x_i) = \mathcal{I}^{in}(x_i) + \sum_{j=1}^{|\hat{\mathcal{S}}|} \eta$, yields the marginal gain:

$$\Delta(x_i \mid \hat{\mathcal{S}}) = \alpha \mathcal{I}^{in}(x_i) + \alpha \sum_{j=1}^{|\hat{\mathcal{S}}|} \eta + \sum_{x_j \in \hat{\mathcal{S}}} g(D(x_i, x_j)). \tag{26}$$

Since all intrinsic terms are non-negative, monotonicity is ensured when $\eta$ is large enough:

$$\eta \geq \frac{1}{\alpha} \max_{x_i, x_j} |g(D(x_i, x_j))|. \tag{27}$$

This constant does not affect greedy decisions (as it shifts all samples equally), so it is omitted in the main text for clarity.

## F. Proof of the Greedy Approximation Guarantee

In this section, we provide a simple proof of the $(1 - 1/e)$ approximation guarantee for the greedy algorithm applied to our unified objective (Theorem 1). The proof directly handles the general case when $f(\varnothing) = 0$. For a more detailed and rigorous derivation, we refer the reader to the original work of Nemhauser et al. (Nemhauser et al., 1978).

*Proof.* We may, without loss of generality, assume $f(\varnothing) = 0$, since adding a constant to $f$ does not change marginal gains nor the greedy solution. Let $\mathcal{S}_t$ be the greedy set after $t$ steps, with $\mathcal{S}_0 = \varnothing$ and $|\mathcal{S}_t| = t$. By submodularity, for any $t$,

$$\begin{aligned} f(\mathcal{S}^*) &\leq f(\mathcal{S}_t) + \sum_{x \in \mathcal{S}^* \setminus \mathcal{S}_t} \Delta(x \mid \mathcal{S}_t) \\ &\leq f(\mathcal{S}_t) + (b - t) \max_{x \in \mathcal{T} \setminus \mathcal{S}_t} \Delta(x \mid \mathcal{S}_t). \end{aligned} \tag{28}$$

By the greedy rule, we can get the maximum of the marginal gains can be expressed as :

$$\max_x \Delta(x \mid \mathcal{S}_t) = f(\mathcal{S}_{t+1}) - f(\mathcal{S}_t), \tag{29}$$

where $\mathcal{S}_{t+1}$ is the next greedy set. Hence, the following inequality can be derived from Eq. 28:

$$f(\mathcal{S}^*) - f(\mathcal{S}_t) \leq (b - t)\big(f(\mathcal{S}_{t+1}) - f(\mathcal{S}_t)\big), \tag{30}$$

Next, rewrite it at iteration $t+1$ as

$$f(\mathcal{S}^*) - f(\mathcal{S}_{t+1}) = \big(f(\mathcal{S}^*) - f(\mathcal{S}_t)\big) - \big(f(\mathcal{S}_{t+1}) - f(\mathcal{S}_t)\big). \tag{31}$$

Substituting the lower bound above yields

$$\begin{aligned} f(\mathcal{S}^*) - f(\mathcal{S}_{t+1}) &\leq \big(f(\mathcal{S}^*) - f(\mathcal{S}_t)\big) - \frac{f(\mathcal{S}^*) - f(\mathcal{S}_t)}{b - t} \\ &= \Big(1 - \frac{1}{b - t}\Big)\big(f(\mathcal{S}^*) - f(\mathcal{S}_t)\big). \end{aligned} \tag{32}$$

For concice, let $\Delta_t = f(\mathcal{S}^*) - f(\mathcal{S}_t)$ denote the optimality gap after $t$ greedy steps, we get the recursive inequality:

$$\Delta_{t+1} \leq \left(1 - \frac{1}{b-t}\right)\Delta_t, \qquad t = 0, 1, \ldots, b-1. \tag{33}$$

Applying Eq. 33 iteratively gives

$$\Delta_b \leq \prod_{t=0}^{b-1}\left(1 - \frac{1}{b-t}\right)\Delta_0 = \prod_{k=1}^{b}\left(1 - \frac{1}{k}\right)\Delta_0, \tag{34}$$

where we substituted $k = b - t$. Each factor satisfies $1 - \frac{1}{k} \leq 1 - \frac{1}{b}$, hence

$$\prod_{k=1}^{b}\left(1 - \frac{1}{k}\right) \leq \left(1 - \frac{1}{b}\right)^b. \tag{35}$$

Using the inequality $\ln(1-x) \leq -x$ for $x \in (0,1)$, we further obtain

$$\left(1 - \frac{1}{b}\right)^b = e^{b\ln(1-\frac{1}{b})} \leq e^{b \cdot (-\frac{1}{b})} = \frac{1}{e}. \tag{36}$$

Combining with Eq. 34–35, we can conclude that

$$f(\mathcal{S}^*) - f(\mathcal{S}_{\text{greedy}}) = \Delta_b \leq \frac{1}{e} f(\mathcal{S}^*), \tag{37}$$

or equivalently,

$$f(\mathcal{S}_{\text{greedy}}) \geq (1 - 1/e) f(\mathcal{S}^*). \tag{38}$$

$\square$

## G. Complexity Analysis of Greedy Selection

To better position our greedy solver within the broader combinatorial optimization literature, we briefly compare its complexity with those of representative exact and local-search approaches designed for the MWCP below.

### G.1. Our Greedy Selection

Let $b = (1-p)N$ denote the number of selected samples, and let $\bar{m}$ denote the average neighborhood size in the sparsified graph. We maintain current scores with a max-heap and update only the neighbors of each newly selected sample using reverse adjacency lists.

The greedy stage has two parts in our practical implementation: (i) heap initialization (to speed up search), which costs $O(N \log N)$, and (ii) $b$ iterative updates. In each iteration, we extract the current best sample from the heap costs $O(\log N)$, and update the affected neighbors costs $O(\bar{m} \log N)$, since only the neighbors of the newly selected sample need score updates. Therefore, each iteration costs:

$$O(N \log N) + O\big(b(1+\bar{m}) \log N\big).$$

Under our graph sparsification scheme, each sample only interacts with samples in its local cluster, so $\bar{m} \approx \frac{N}{K} \gg 1$, where $K$ is the number of clusters. This gives the following approximation of the above equation:

$$O\big(N \log N + b(1+\bar{m}) \log N\big) \approx O(N \log N),$$

which is close to sorting-like complexity in practice and substantially cheaper than recomputing pairwise gains over all remaining samples at every step.

### G.2. Exact Branch-and-bound Solver

We also provide the complexity analysis of an exact branch-and-bound solver (Babel, 1994), which uses weighted coloring to compute upper bounds and guide branching. In general, its runtime can be written as $O(BN^2)$, where $B$ is the number of visited branch-and-bound nodes and the $O(N^2)$ term reflects repeated weighted-coloring and pruning operations at each node. Since $B$ is exponential in the worst case, the overall complexity remains exponential.

In our setting, each subgraph is fully connected, so clique feasibility no longer provides strong pruning, and the problem reduces to selecting a fixed-size subset of size $b = (1 - p)N$. This solver therefore essentially searches over size-$b$ subsets. By Stirling's approximation,

$$\binom{N}{(1-p)N} \approx e^{NH(1-p)},$$

where $H(q) = -q \ln q - (1 - q) \ln(1 - q)$ is the binary entropy function. This is the standard asymptotic approximation for binomial coefficients when the subset size is a fixed fraction of $N$, and therefore gives a natural complexity estimate here. Hence, the worst-case complexity is on the order of $O\left(e^{NH(1-p)}N^2\right)$.

Even if the same sparsification is applied so that each cluster has size about $N/K$, the complexity only reduces to approximately

$$O\left(\frac{N^2}{K}e^{\frac{N}{K}H(1-p)}\right) \approx O\left(e^{NH(1-p)}N^2\right),$$

which still remains exponential. This exponential complexity is unacceptable for large datasets with millions samples.

### G.3. Heuristic Local Search Solver

We also analysis the complexity of a heuristic phased local search method (Pullan, 2008), which repeatedly $T$ times of add/swap/perturb op. on the current clique. In our fixed-size setting, a natural budget is $T = O(b)$, which gives an overall complexity of about

$$O(Nb^2) = O\left((1-p)^2 N^3\right),$$

which is roughly $O(N^3)$.

### G.4. Summary

For clarity, we compare the complexity of these methods in Tab. 10. This complexity comparisons suggest that exact branch-and-bound methods remain exponentially expensive, and local-search heuristics also introduce substantially higher-order cost. By contrast, our method remains scalable while preserving the unified objective framework.

*Table 10.* Comparison of methods and their complexity.

| Method | Type | Complexity in our setting |
|---|---|---|
| Ours (Algorithm 1) | Greedy | $O(N \log N)$ |
| (Babel, 1994) | Branch-and-bound | $O\left(e^{NH(1-p)}N^2\right)$ |
| (Pullan, 2008) | Local-search | $O(N^3)$ under $T = O(b)$ |

## H. More Implementation Details

Most classification experiments are conducted on an 8×A800 GPU server. Runtime experiments in Tab. 3 of the main text are additionally conducted on a 4×4090 server to reflect realistic training scenarios. Please note that the running time is also related to the CPUs to perform graph sparsification and greedy selection. Our server is equipped with two Intel Xeon Platinum 8468V CPUs. For CIFAR-10/100 and ImageNet-1k, we follow InfoBatch (Qin et al., 2023) training settings; for Swin-T we follow Dyn-Unc (He et al., 2024). AutoAugment (Cubuk et al., 2018), random path drop, and gradient clipping are applied only to Swin-T to ensure a fair comparison with Dyn-Unc. The hyperparameters required for reproduction are listed in the Tab. 11. We use $a/b/c$ to denote different choice for 30%, 50%, 70% pruning ratios. For the fairness of comparison, the reported performance in Tab. 1 and Tab. 2 are implemented with entropy as the intrinsic term, $g_1$ and cosine

distance for extrinsic term. *However, please note that on large prune ratios, such as 70%, using loss values as intrinsic term usually leads to a better performance.* This phenomenon is also described in the Tab. 5. We believe for large prune ratios, loss values can identify those samples which can offer more information model still have not mastered.

*Table 11.* Detailed training settings on classification datasets.

| | Parameters | CIFAR-10 | CIFAR-100 | ImageNet-1k | |
|---|---|---|---|---|---|
| | Models | ResNet-18 | ResNet-18 | ResNet-50 | Swin-T |
| Training | optimizer | SGD | SGD | Lars | AdamW |
| | weight_decay | 0.0005 | 0.0005 | 0.00005 | 0.05 |
| | batch_size | 128 | 128 | 1024 | 1024 |
| | epochs | 200 | 200 | 90 | 300 |
| | learning_rate | 0.10 | 0.05 | 6.4/6.4/3.6 | 0.001 |
| | label smoothing | 0.1 | 0.1 | 0.1 | 0.1 |
| | learning rate scheduler | OneCycle | OneCycle | OneCycle | CosineAnnealing |
| | learning rate warmup | - | - | 5 | 20 |
| Pruning | $\alpha$ | 1.0 | 1.0 | 1.0/0.8/0.2 | 1.0 |
| | epoch start pruning | 5 | 5 | 2 | 2 |
| | epoch stop pruning | 180 | 180 | 80 | 265 |
| | feature for graph building | Well Trained | Well Trained | Torch Pretrained | Torch Pretrained |

