# OpenReview forum: "Selecting Samples on Graphs: A Unified Dataset Pruning Framework for Lossless Training Acceleration"
_ICML.cc/2026/Conference — ICML 2026 regular_

### Official Review · Reviewer_QSiU · 2026-03-11

**Soundness:** 3
**Presentation:** 3
**Significance:** 3
**Originality:** 3
**Overall Recommendation:** 3
**Confidence:** 4

**Summary:**

The authors propose UGIES, a novel graph-based framework that elegantly unifies intrinsic (sample difficulty/informativeness) and extrinsic (redundancy/diversity) importance criteria. By framing subset selection as a Maximum Weight Clique Problem (MWCP), the method leverages a greedy algorithm with a theoretical submodularity guarantee under mild conditions. Structured graph sparsification is introduced to ensure scalability.

**Compliance With Llm Reviewing Policy:**

Affirmed.

**Final Justification:**

After reading the rebuttal, I confirm the proposed pipeline introduces more computations compared than the standard sorting baseline.

Although the overhead is small compared to the full pretraining process, it's large (8 times) compared to the time required for the standard sorting baseline.

I will keep the original rating.

**Key Questions For Authors:**

Q1: Does the pairwise assumption over-penalize dense clusters? For redundant samples {A,B,C}, the pairwise sum AB and AC  penalizes A twice, failing to capture redundancy saturation. This linear accumulation ignores the submodular nature of information overlap, leading to potential distribution distortion.



Q2: The structured graph sparsification ignores the redundancy among classes. Take ImageNet-1K as examples, class (white shark) and class (harmmerhead shark) have very high feature similarity.



Q3: Following Q2, the ablatioin of Structured Graph Sparsification can be added. While MWCP is intractable without greedy selection approximation, a result with no sparsification on small-scale datasets (i.e., CIFAR) can be provided to see the impact on both efficiency and performance.



Q4: Will a “DYNAMIC” coefficient alpha help? Since the score is computed epoch-wise, and the intrinsic importance is highly related to the learning dynamics. Intrinsic importance is more likely to be useful in the early training stage.



Q5: The choice of EL2N-50 in Table 3 needs further clarification. In other works (e.g., CCS [1]) uses first 10 epochs’ L2 prediction error, which better aligns EL2N’s own claim on early training epochs.

[1] "Coverage-centric coreset selection for high pruning rates."  ICLR’23.

**Limitations:**

see weaknesses

**Strengths And Weaknesses:**

## Strengths
1. The paper provides solid theoretical grounding (Lemma 3.4, Theorem 3.5), proving that the unified objective is submodular under interpretable conditions, which guarantees the approximation bounds of the greedy solver.

2. Strong Empirical Performance: The method demonstrates impressive lossless compression capabilities (e.g., 50% pruning on ImageNet) and clearly outperforms recent competitive baselines (e.g., InfoBatch, DivBS).


## Weaknesses
there is a critical missing piece regarding the computational overhead analysis:

1. **Lack of explicit comparison between Sorting and Graph Construction costs**: Standard intrinsic pruning methods essentially require computing scalar scores and applying a simple sorting operation. In contrast, the proposed method requires building a graph (computing pairwise distances) and running a greedy selection algorithm.

2. Although Section 3.5 introduces structured graph sparsification to avoid the worst-case $O(N^2)$ complexity, and Table 3 provides an aggregate "preprocess" time (0.55h), the manuscript does not clearly compare the theoretical complexity of these two paradigms.

3. The authors need to explicitly outline the time and space complexity of the proposed pipeline (clustering + intra-cluster distance computation + greedy search) versus the standard sorting baseline. Furthermore, it would be highly beneficial to break down the "preprocess" overhead in Table 3 into sub-components (e.g., feature extraction vs. clustering vs. graph building vs. subset selection) so readers can clearly see where the computational bottlenecks lie and how they scale.

---

> ### Author Rebuttal · Authors · 2026-03-27
>
> We sincerely thank the reviewer for the recognition of our theoretical grounding, performance, and originality.
>
> ### (W1-3) Complexity and Time
>
> **Complexity.**
> For intrinsic-only pruning, the selection cost is sorting only, i.e., $O(N\log N)$ time and $O(N)$ memory.
>
> For UGIES, let $K$ be the total number of clusters, $C$ the number of classes, $d$ the feature dimension, $t$ the number of k-means iterations, and $b=(1-p)N$ the retained subset size. On ImageNet-1k, $N=1.2e6,C=1e3,K=1.6e3,t=10,d=256$. We use P-way parallelism with $P=32$.
>
> Graph construction has two parts:
> - Clustering. Each class has about $N/C$ samples and is split into $K/C$ clusters, so the time complexity under P-way parallelism is $O(\frac{C}{P}\cdot \frac{N}{C}\cdot \frac{K}{C}\cdot t d)$, which is nearly $O(N)$ since $\frac{Ktd}{CP} \ll N$ .
> - Intra-cluster distances. The average cluster size is $\bar n\approx N/K$. Computing pairwise distances within a cluster costs $O(\bar n^2 d)$, and the time complexity over all clusters under P-way parallelism is $O(\frac{N^2 d}{KP})$.
>
> For greedy selection, with reverse adjacency lists and a max-heap, only neighbors of the newly selected node are updated. Let $\bar m\approx N/K$ be the average neighborhood size. Then the selection cost is $O((N+b\bar m)\log N) \approx O(N\log N)$ since $\bar m\ll N$, i.e., roughly the same order as sorting.
>
> Thus, the overall complexity of UGIES is
> $O(\frac{NKtd}{CP}+\frac{N^2 d}{KP}+(N+b\bar m)\log N)$,
> with the dominant term being sparse graph construction $O(N^2 d/KP)$.
>
> For memory, sorting needs $O(N)$. UGIES stores sample features and sparse intra-cluster distances, giving
> $O(Nd+N^2/K)$
> if all sparse edges are materialized.
>
> **Wall-clock breakdown.**
> Please note that greedy subset selection is exactly the reported **pruning** time 0.07h, not part of preprocess (prep.). Feature extraction takes **5 min** with FP16 GPU inference; clustering takes **23 min** on CPU with FAISS; intra-cluster distance computation and graph instantiation take **5 min** with GPU + multithreading. Thus prep. totals **33 min (0.55h)** in Tab.3. The main bottleneck is CPU clustering; our machine uses 2 × Intel Xeon Platinum 8380C.
>
> ---
> ### (Q1) Redundancy saturation
>
> We'd like to clarify that UGIES does not use a static sum to rank samples independently. The selection is based on the dynamic **set-dependent marginal gain** in Eq.(8): the importance varies along with more samples are added into $S$. Once similar samples have already been selected, the marginal utility of adding another redundant sample naturally decreases. Therefore, in Alg.1, redundancy is not penalized in a “double-counting” manner, but through diminishing incremental benefit conditioned on the current subset.
>
> ---
> ### (Q2 & Q3) Cross-class redundancy and sparsification ablation
>
> Thank you for these insightful comments. UGIES is a flexible framework that can naturally incorporate cross-class information. We can add an extra term to the extrinsic importance to better preserve samples that are highly similar to clusters from other classes, so that boundary samples between highly similar and confusing classes are more likely to be preserved. For example, we can define
> $\mathcal{I}^{ex}_{cross}(x_i|\mathcal{S})=\max_c \left[-\phi \left(D^{cos}(x_i,c)\right)-1\right]$,
> where $c$ denotes a cluster center in the set of all cluster centers from other classes. This reflects the flexibility and extensibility of our unified framework.
>
> We understand Q3 as asking whether ignoring cross-class similarity affects performance. This can be tested **without giving up the sparsification speedup**. On ImageNet-1k with ResNet-50 at 50% pruning, we compare:
> - A) random partition instead of class-wise;
> - B) our current class-wise partition;
> - C) current + the above term $\mathcal{I}^{ex}_{cross}$ (with weight 0.2).
>
> The results suggest that random partition degrades performance, while adding cross-class information gives a further, thouh small, improvement. This supports the effectiveness of the current sparsification, while also showing that cross-class redundancy is a promising extension that UGIES can naturally support.
>
> |Baseline|A(Random)|B(Ours)|C(+Cross term)|D(Dynamic α)|
> |:-:|:-:|:-:|:-:|:-:|
> |76.35|76.08|76.46|76.54|76.19|
>
> ### (Q4) Dynamic α
>
> Dynamic α as an interesting extension for future work. Following your valuable advice, we ran a simple test at 50% pruning by using 2α in the early stage (setting D above). D suffers from a decrease of Acc compared with B(Ours). A possible reason is that overly focusing on intrinsic imp. may lead to poor diversity and hurt the generalization.
>
> ---
> ### (Q5) EL2N in Table 3
>
> We first tried EL2N-20 under 30% pruning, but only obtained 62.7 Acc, which we consider too poor. We found that the per-sample error changes dramatically in the first 15 epochs, so we switched to EL2N-50 to obtain more stable signals and a better performance. We will clarify this in the revision.

---

> > ### Author Rebuttal · Reviewer_QSiU · 2026-04-03
> >
> > 1. In stead of using paragraphs, please use table to clearly compare Complexity, Time etc for the proposed pipeline (clustering + intra-cluster distance computation + greedy search) versus the standard sorting baseline. Which one is higher? To which degree?
> >
> > 2.  On Q1: Structural Bias of $\mathcal{I}^{ex}$.
> >
> > The rebuttal focuses on the greedy selection procedure, but our concern is with the functional form of $\mathcal{I}^{ex}$ itself, which is defined as a linear sum of pairwise penalties:
> >
> > $\mathcal{I}^{ex}(x_i \mid S) = \sum_{x_j \in S} g\big(D(x_i, x_j)\big)$
> >
> > This formulation introduces a potential structural bias:
> >
> > - A sample near a large redundant cluster (e.g., many highly similar points) accumulates a linearly increasing penalty
> > - Even when those points carry overlapping information, leading to over-penalization
> >
> > Two concrete numerical examples from real data would be useful:
> >
> > - (a) a sample near a large redundant cluster
> >
> > - (b) a sample near a small but diverse cluster
> >
> > Please report the actual accumulated penalty values, so that the magnitude of this bias can be directly assessed.
> >
> >
> > 3. **On Q3**,  the Effect of Graph Sparsification.
> >
> > The rebuttal compares random vs. class-wise partitioning. But an ablation that directly evaluates the impact of graph sparsification is good.
> >
> > - Compare **with** vs. **without** sparsification on a small dataset (e.g., CIFAR), where dense computation is feasible
> > - Report the accuracy-efficiency trade-off

---

> > > ### Author Response · Authors · 2026-04-04
> > >
> > > **W1**: Following your advice, we provide the comparison below:
> > > |Complexity|Cluster|Distance|Greedy|Total|
> > > |-|-|-|-|-:|
> > > |Ours|$O(N)$|$O(\frac{N^2 d}{KP})$|$O(N\log N)$|$O(N+N^2+N\log N)$|
> > > |Sorting|-|-|$O(N\log N)$|$O(N\log N)$|
> > >
> > > |Run Time|Cluster|Distance|Greedy/Sort|total|
> > > |-|-|-|-|-|
> > > |Ours|23m (CPU)|5.3m (GPU)|4.4m|32.7m|
> > > |Sorting|-|-|4.2m|4.2m|
> > >
> > > Overall, our pipeline introduces an additional $O(N+N^2)$ term compared with the standard sorting baseline. However, because the distance computation is GPU-accelerated, the extra time on ImageNet is less than half an hour in practice, and most of this overhead comes from clustering on CPU. Note that the clustering and distance comput. results can be reused multiple times for different pruning ratios, so this overheads is acceptable in practice. Moreover, compared with the 13.9h required for one full-data training run, this overhead is relatively small **(only 3.9% of full training)**.
> > >
> > > **Q1**: Thank you for giving us a second chance to discuss this question. The reason we emphasized the greedy selection process in our previous response is that the penalty here is explicitly depends on the current subset $S$: it is related to  which samples have already been selected into $S$, and among them, which ones also have edges connected to $x_i$ (within the same cluster).
> > >
> > > Hence, our previous discussion of the greedy process was not meant to avoid your concern about the functional form itself. Rather, we intended to clarify how the penalty is actually formed during selection. For example, if the total size of the cluster containing a given sample is 50, but only 20 of them have currently entered $S$, then the extrinsic term for this sample only contains the 20 linearly accumulated terms corresponding to these 20 connected neighbors, rather than all 50 samples in the cluster. This dynamic mechanism natually reduce the number of linear addition term and avoid uncontrolled over-penalization.
> > >
> > > To directly address your concern, we further provide numerical values for two example samples from real data. We extracted two samples with similar intrinsic importance during the first pruning step before training on CIFAR-100, and performed 20% pruning (i.e., retaining 80% of the data, corresponding to 40k samples).
> > >
> > > |Case|Sample|Cluster Size|Cluster Mean Dist.|$I^{ex}$(S=0)|$I^{ex}$(S=20k)|$I^{ex}$(S=40k)|$I^{in}$ (CE Loss)|Cluster Mean $I^{in}$|
> > > |-|-|-|-|-|-|-|-|-|
> > > |a|$x_1$|39|0.1339|0|-4.8227 (12 neighbors in $S$)|-9.0736 (22 neighbors in $S$)|4.27|4.31|
> > > |b|$x_2$|21|0.3531|0|-4.0856 (9 neighbors in $S$)|-7.1429 (18 neighbors in $S$)|4.33|4.01|
> > >
> > > As shown above, the gap between these two cases is not large. First, due to this dynamic mechanism, not all samples in a large cluster linearly contribute to the penalty term. Since larger clusters tend to be more redundant, they are also pruned more aggressively under the penalty term, which in turn limits the actual number of selected neighbors that participate in the accumulation. Second, this is also helped by the fact that, in our actual clustering procedure, we impose a soft constraint on cluster size so as to obtain clusters with relatively balanced sizes. Under this setting, the size differences across clusters are not large to begin with. For example, when $K=16$, using the above procedure on CIFAR-100, the minimum and maximum cluster sizes within the same class are 21 and 39, respectively, as shown above.
> > >
> > > **Q3**: Thank you for this helpful clarification, and we apologize for the misunderstanding the focus of your question. We conducted an ablation study on graph sparsification (GS) on CIFAR-100. To obtain the results quickly, we used a machine with 2 x Intel Xeon Platinum 8380C CPU + 1 x H800 GPU (5 runs). The reported Graph Building Time includes clustering, distance calculation, and graph construction. Since the feature extraction time is the same in both settings, we do not include it here for a clearer comparison.
> > >
> > > |Acc |w/ GS|w/o GS|
> > > |-|-|-|
> > > |30%| 78.9 | 78.8 |
> > > |50%|78.6| 78.3
> > > |70%|77.6| 77.4 |
> > >
> > > |Time(s) p=50%|w/ GS|w/o GS|
> > > |-|-|-|
> > > |Graph Building|75.31s|928.17s|
> > > |Greedy Selection|22.26s|25.64s|
> > > |Training|454.32|455.41s|
> > >
> > > The phenomenon shown in the two tables above is quite interesting. After completely removing the class-wise and cluster-wise grouping, the accuracy of w/o GS actually drops slightly. We believe this further highlights the importance of preserving the two-level structure of class-level partitioning and cluster-level pruning. Without the separation by class and cluster, samples from different classes but with similar features may be incorrectly treated as redundant to each other. It may also prevent the pruning procedure from capturing the fine-grained representative semantic structure at the cluster level.
> > >
> > > In terms of efficiency, w/o GS is clearly much slower. This is because both the number of pairwise distance computations and the number of constructed edges increase substantially compared with w/ GS.

---

### Official Review · Reviewer_bkLR · 2026-03-13

**Soundness:** 3
**Presentation:** 3
**Significance:** 2
**Originality:** 3
**Overall Recommendation:** 4
**Confidence:** 4

**Summary:**

The paper proposes a data pruning framework by modeling the dataset as a weighted graph and treating pruning as a Maximum Weight Clique Problem. The paper shows that the proposed method has a formal approximation guarantee under some conditions. The actual selection uses a greedy solution and is evaluated extensively.

**Compliance With Llm Reviewing Policy:**

Affirmed.

**Final Justification:**

The rebuttal addressed my original concerns and reinforced my prior assessment regarding the soundness and originality of the paper. However, the complexity and overhead issue raised by another reviewer has also been brought to my attention. Overall, I think the paper proposes a novel framework with pruning quality improvement at the cost of some overhead, and I will retain the weak accept rating.

**Key Questions For Authors:**

1. How does the proposed method work in imbalanced scenarios or fine-grained datasets?

2. How to select $\alpha$ for any dataset?

3. Is there enough discussion on prior works that consider a graph solution?

**Limitations:**

yes

**Strengths And Weaknesses:**

Strengths:

1. To my knowledge, the paper is technically sound. The paper provides necessary information regarding the assumptions for important claims, such as submodularity and approximation guarantee. Although I did not check everything in detail, the proof sketch makes sense to me.

2. The presentation quality of the paper is above average. All sections are clearly organized and the results are presented nicely.

3. To my knowledge, the paper is original because of the novel solution to data pruning through modeling the dataset as a graph problem. The idea is a natural one for combining the intrinsic and extrinsic signals for data contribution. The concern about using a graph method would be the complexity, but the paper uses a greedy strategy and demonstrates low computational cost compared to baseline methods.

4. Considering the above, I think the paper makes a decent contribution to the related topic.



Weaknesses:

1. The main comparisons are for CIFAR-10/100 and ImageNet-1k, which are all balanced image datasets. There’s not too much variety and proof of generalization ability of the proposed method.

2. The balancing parameter $\alpha$ still seems difficult to adjust in real applications.

---

> ### Author Rebuttal · Authors · 2026-03-26
>
> ## Response to Reviewer bkLR
>
> We sincerely thank the reviewer for the positive assessment of our paper’s soundness, originality, presentation quality, and practical efficiency. We are especially encouraged that the reviewer finds the method technically sound and the graph-based formulation a natural way to unify intrinsic and extrinsic signals. We also appreciate the reviewer’s thoughtful questions on generalization, practical tuning, and related work, which help us improve the manuscript.
>
> ---
> ### (Q1&W1) Results on imbalanced and fine-grained datasets
>
> Thank you for raising this important point. We would like to respectfully clarify that such experiments are already included in the Appendix Sec. E.
>
> Beyond the CIFAR and ImageNet in the main text, we evaluate UGIES on **PASCAL VOC 2012 semantic segmentation** and **COCO 2017 object detection** in Sec. E.1/E.2, which are much more relevant to the reviewer’s concern on imbalance and long-tail behavior. These tasks naturally involve class imbalance and, especially for COCO 2017, pronounced long-tail distribution. On PASCAL VOC 2012, our method achieves the best mIoU at all aggressive pruning ratios(70%, 80%, 90%), outperforming SOTA methods DivBS and InfoBatch. On COCO 2017, UGIES also achieves slightly better AP than the baseline across all reported metrics at 25% pruning. These results show that UGIES generalizes well beyond balanced datasets.
>
> More broadly, our formulation is not depend on balanced label distributions: intrinsic importance measures sample informativeness, while extrinsic importance suppresses redundancy through pairwise relations. This is particularly helpful under imbalance, where one wants to retain informative minority samples while avoiding over-selecting redundant majority samples.
>
> Regarding fine-grained scenarios, ImageNet-1k contains many visually similar classes and thus serves as a meaningful proxy. More importantly, UGIES operates on the sample-level and there pairwise signals so it is capable of distinguishing similar categories and naturally applicable to fine-grained datasets as well. We will make this broader applicability more explicit in the revision.
>
> ---
> ### (Q2&W2) How to select α and its difficulty
>
> Thank you for this insightful and practical question. We would like to clarify that **α is not a black-box coefficient** in UGIES. It has a clear interpretation: it controls the trade-off between intrinsic and extrinsic importance. Our ablation on α already provides a practical guideline. As discussed around Fig. 1 and Lines 409–419, no single fixed α is optimal across all pruning ratios: larger α works better when pruning is mild, while smaller α becomes preferable under more aggressive pruning, where diversity and redundancy control matter more.
>
> This leads to a simple rule of thumb: use smaller α for higher pruning ratios or more redundant samples(like COCO with a great amount of human instances), and use larger α when the data is already diverse and one wishes to emphasize harder or more informative samples. In this sense, UGIES does not merely introduce a tunable parameter; it provides an **interpretable "knob"** together with an **empirical principle** for setting it. We agree that such guidance should be stated more explicitly, and we will revise the paper to highlight this practical guideline more clearly.
>
> ---
> ### (Q3) Discussion on graph-based works
>
> Thank you for pointing this out. We agree that the discussion of prior graph-related works can be strengthened. In the current manuscript, the most relevant prior methods are the recent hybrid methods discussed in Sec. 2, especially D²-pruning and InfoMax, since they also combine instance-level and pairwise signals. Our intention was to position UGIES against such fixed hybrid metrics by emphasizing that UGIES provides a unified objective family, an objective-derived greedy solver, and general design conditions for submodularity and approximation guarantees, rather than offering one specific hybrid metric.
>
> Moreover, despite that there are some graph solution for the GNN training (graph neural network) and feature selection (keeps all data but filter some feature dimentions of all data), they focus on completely different aspects with data pruning.
> We could include more discussion with these works in the future.

---

> > ### Author Rebuttal · Reviewer_bkLR · 2026-04-03
> >
> > I agree with some of the concerns raised by other reviewers, but also think that they are adequately addressed by the author rebuttal. Thus, I tend to maintain my rating.

---

> > > ### Author Response · Authors · 2026-04-07
> > >
> > > We sincerely thank you for your positive feedback and for considering our previous rebuttal helpful.
> > >
> > > We have now provided additional clarifications in response to the other reviewers’ comments, further elaborating on several key concerns. We would greatly appreciate it if you could kindly take a look at these updated responses, as they may also help address the concerns you mentioned.
> > >
> > > If you find that the additional clarifications have sufficiently resolved the remaining issues, we would be truly grateful if you could consider reflecting this in your overall assessment.
> > >
> > > Thank you again for your time and thoughtful evaluation.

---

### Official Review · Reviewer_aeuM · 2026-03-19

**Soundness:** 3
**Presentation:** 3
**Significance:** 2
**Originality:** 2
**Overall Recommendation:** 5
**Confidence:** 3

**Summary:**

This paper introduces UGIES, a dataset pruning framework that models training samples as a weighted graph where each node's degree serves as an intrinsic importance measure, while edge weights capture extrinsic importance via a transformation function $g$ reflecting the information contributed by neighboring samples.

The pruning problem is formulated as a Maximum Weight Clique Problem (MWCP): finding a clique of size $b$ that maximizes total importance. The objective is naturally NP-hard. Under specific conditions on the pairwise distances and the interaction mapping $g$, the objective is submodular, enabling the classical $(1-1/e)$ greedy approximation guarantee from Nemhauser et al. (1978).

Experiments span CIFAR-10/100, ImageNet, and detection tasks, showing competitive pruning performance with practical wall-clock savings (up to ~40% reduction).

**Compliance With Llm Reviewing Policy:**

Affirmed.

**Final Justification:**

The main concern was about actual performance and the incremental nature that adopts a "unifying framework" approach.
The "unification" actually yields competitive performance without being too specific on the choice of intrinsic/extrinsic metrics, which might open up better possibilities by leveraging more powerful graph learning approaches.
So we're raising to weak accept for the versatility until more analysis is done on the the MWCP part.

**Key Questions For Authors:**

- What is the concrete theoretical or empirical advantage of UGIES over CRAIG beyond the graph-based presentation? A direct comparison and discussion would be informative.
- Given the 1.1% accuracy drop at 70% pruning on ImageNet, can you justify the "Lossless" claim in the title, or would a more qualified statement be appropriate?
- The margins over baselines are often within 0.1-0.5%. Have the authors conducted multiple runs with different seeds? Can confidence intervals or significance tests be provided?

**Limitations:**

Yes.

**Strengths And Weaknesses:**

The core of the algorithm relies on Lemma 3.4 (submodularity): since $g$ maps non-negative values to non-positive, adding elements contributes non-positive marginal terms, yielding diminishing returns. The monotonicity is addressed in Appendix F via uniform additive shift, which should not affect greedy decisions. Theorem 3.5 is a direct and correct application of the classical Nemhauser et al. (1978) guarantee.

The MWCP framing can be misleading. On a fully connected graph, every subset is trivially a clique. Thus, the "clique" constraint is redundant and the problem reduces to standard cardinality-constrained submodular maximization. The emphasis on the "NP-hard MWCP" narrative overstates the difficulty of the actual optimization in such case. Algorithm 1 has minor errors: Line 234 is missing the $g$ mapping (should be $g(D(x_i, x^*))$, and Line 235 is missing the $\alpha$ coefficient from Eq. (4). Eq. (4) is double-counting the edges compared to Eq. (3) (assuming undirected graph) . This doesn't affect optimization, this can be simplified by adding the $1/2$ factor commonly used in graph cuts.

The paper is generally well-written and a pleasure to read. The unified framework combining intrinsic and extrinsic signals into a graph formulation is clearly explained. Comprehensive experiments across datasets (CIFAR-10/100, ImageNet), architectures (ResNet, Swin-T), and tasks (classification, detection).

While the title claims "Lossless" training acceleration, but results show a 1.1% accuracy drop at 70% pruning on ImageNet (this overstates the contribution). There are some tiny citations formatting like "Tan & et.al, 2025" appearing multiple times (Lines 64, 97). InfoBatch is attributed to Xia et al. (2022) instead of Qin et al. (2023) at table 2 on Swin-T. A Stray "ok" in line 181-182 (first column), typos: "controll" L152, "limiation" L437.

The practical wall-clock savings (40% reduction) are demonstrated and meaningful. The modular plug-in design accommodating diverse intrinsic/extrinsic metrics is useful for practitioners. Howevew, the margins over strong baselines are small (often 0.1-0.5%), with no significance tests or confidence intervals reported, and the Swin-T baseline accuracy (79.6%) is lower than commonly reported (~81%), which may inflate the relative improvement.

The explicit MWCP formulation with separate node/edge weights is a clean way to unify intrinsic and extrinsic importance signals. The structured graph sparsification via class-then-cluster neighborhoods is a practical design choice, although a common practice in graph learning.

There is a substantial overlap with CRAIG (Mirzasoleiman et al., 2020), which also uses submodular data selection with the same $(1-1/e)$ guarantee. CRAIG is cited in Table 1 but insufficiently discussed in the related work despite being the most similar prior work. The submodularity proof is straightforward, and combining intrinsic (importance) + extrinsic (diversity) signals was explored by papers such as InfoMax.

Overall, the theoretical novelty seems overstated: the submodularity guarantee is standard, and the work is insufficiently differentiated from prior submodular data selection methods (particularly CRAIG). Empirical margins are small and lack statistical validation and the "lossless" claim is not supported at high pruning ratios. The contribution amounts to an interesting graph-theoretic reformulation of an existing problem without substantial additions beyond that.

---

> ### Author Rebuttal · Authors · 2026-03-24
>
> ## Response to Reviewer aeuM
>
> We thank the reviewer for the careful reading and positive feedback on clarity and efficiency. We respond to each point below.
>
> ---
>
> ### (Q1) Relation to CRAIG and advantage
>
> **Difference.**
> We have to respectfully argue that UGIES and CRAIG optimize fundamentally different objectives.
> CRAIG selects a weighted subset to approximate the **full-data gradient** to match with the full-data training dynamics.  UGIES instead formulates pruning as a **unified in/extrinsic importance maximization**.
> UGIES is also not tied to a single objective: it defines a **general objective family** over diverse in/extrinsic metrics, and gives mild conditions for submodularity and greedy approximation. This shifts the contribution from optimizing one objective to providing **design principles**.
>
> **Empirical advantage.**  Since CRAIG aims to approximate full-data behavior, its performance is inherently limited to that of the original dataset. When redundancy and noise exists, as evidenced by our results where removing 30%–50% of data still matches or improves Acc in Tab.1 and 2, approximating full-data gradients may inherit such low effective and even harmful information.
> UGIES instead balances informativeness and redundancy, effectively optimizing the data distribution, enabling less redundancy and more informative subsets, and thus may exceeding full-data performance.
>
> ---
>
> ### (Q2) The term "lossless"
>
> In prior work (e.g., InfoBatch:  **Lossless** training speedup), "lossless" refers to **practical regimes with negligible/no drop**, rather than all pruning ratios.
> Similarly, we achieve lossless performance at 50% and 30% pruning on ImageNet-1k with ResNet-50 and Swin-T, respectively.
> At higher pruning (e.g., 70%), a small drop (~1%) is expected, yet UGIES still outperforms prior works (including InfoBatch).  Thus, our usage follows established convention and is supported by results.
>
> ---
>
> ### (Q3) Statistical significance
>
> All results and error bars (Tab.1,2,9) of our method and the baseline are from **5 independent runs** with random seeds.
> Below we report 95% CI and paired t-test at 30% pruning (ResNet-50, ImageNet).
>
> |Method|Mean|Std| CI(95%)|||5-Run||||
> |:-:|:-:|:-:|:-:|:-:|:-:|:-:|:-:|:-:|:-:|
> |Full-data|76.35|0.15|[76.17, 76.53]|76.35|76.18|76.43|76.24|76.55|
> |Ours|76.96|0.09|[76.86, 77.08]|77.08|77.02|76.83|76.98|76.93|
>
> The CIs do not overlap. Paired t-test gives **p = 0.0028**, indicating statistical significance.
> We will clarify run count and CI in the revision.
>
> ---
> ## Other concerns
> ### 1. MWCP formulation
>
> We respectfully disagree that the MWCP framing is misleading and overstates difficulty.
>
> Even without the graph formulation, selecting a subset under a sample constraint is also **NP-hard**, widely recognized in data selection/coreset literature. This stems from the exponential subset space and does not depend on the MWCP form.
>
> The graph view is not cosmetic. Our extrinsic imp. is explicitly defined via edge weights, making the node-edge representation a natural and transparent way to express the objective. Moreover, the MWCP formulation provides a unified way to capture in/extrinsic imp., highlight the set-dependent nature of the objective, and motivate the resulting greedy rule and metric design guidelines.
>
> In Sec. 3.5 Line 244-255, we also have clarified that the MWCP also remain valid without modification for graphs after the graph sparsification. The sparisified graph is not full connected in practice. However, to accomodate both dense and sparse graphs, we use a more unified version (fully connected graph) in Definition3.1 (Eq. 3).
>
> We will clarify this and look forward to discuss the formulations with you in the disscussion phase. Your advice will be very helpful for improving our manuscript.
>
> ---
>
> ### 2. Swin-T results
>
> This concern likely comes from comparing different training recipes.
> The ~81% result typically uses EMA or stronger settings.
> However, our Swin-T:
> - follows **Dyn-Unc protocol** (Line-883 Sec.H)
> - uses **no EMA or powerful tricks**.
>
> Under this setup, the full-data baseline is **79.6**, and all methods are compared fairly. We also tested an EMA-based setup under 30% pruning ratio. Our conclusion remains unchanged.
>
> |Full-data|Ours|InfoBatch|Dyn-Unc|
> |-|-|-|-|
> |80.6±0.1|**81.0±0.1**|80.6±0.1|80.8±0.1|
>
> ---
>
> ### 3. Comparison with InfoMax
>
> As noted in Sec.2 (Line 91-109 right column), we agree prior work, including InfoMax, combines instance-level and pairwise signals.
> While InfoMax focus on a single fixed metric, our contribution is to elevate this into a **unified framework** that:
> - subsumes a broad family and derives greedy selection of various such designs,
> - and provides general design conditions with approximation guarantees.
>
> ---
>
> ### 4. Minor issues
>
> We thank the reviewer for the careful review and will fix them in the revised version. But they do not affect the method or our main conclusions.

---

> > ### Author Rebuttal · Reviewer_aeuM · 2026-04-03
> >
> > We thank the authors for their thorough response. We are raising our score to a weak accept, conditioned on the paper making its relationship to prior work leveraging sub-modularity more explicit, so that the unifying nature of the proposed framework is better emphasized.
> > Our initial reading led us to interpret the primary contribution as achieving superior performance based on the "lossless" claims; however, the authors' response has clarified that the core contribution lies in introducing a more versatile and general approach, and the added results show that the unified framework is competitive with prior work.
> >
> > This perspective could be further strengthened by including a discussion of more effective algorithms for the Maximum Weight Clique Problem beyond the current greedy method, for instance, exact branch-and-bound solvers or local search heuristics that have demonstrated strong performance in the combinatorial optimization literature.
> >
> > We are willing to raise our score to a full accept should the authors explore, either theoretically or experimentally, such alternative approaches for MWCP and demonstrate their potential within the proposed framework.

---

> > > ### Author Response · Authors · 2026-04-05
> > >
> > > We sincerely thank you for the time and effort on our work. We are especially grateful for your positive assessment of our rebuttal and for recognizing the flexibility and unifying nature.
> > >
> > > Compared with prior methods that exploit submodularity, such as CRAIG, our contribution is not a single specific metric, but rather a more general family of objectives together with the corresponding greedy solution framework, as well as the conditions under which greedy optimization enjoys convergence guarantees and can be used efficiently for pruning. In this sense, as long as a metric satisfies these conditions, or can be reformulated to satisfy them, our framework can also incorporate metrics proposed by prior works. We will revise Sec. 2 or the Appendix to more clearly discuss the relationship between our framework and prior submodularity-based methods so that the unifying perspective is better highlighted.
> > >
> > > ## Complexity and time comparison with representative MWCP solvers
> > >
> > > Your suggestion to discuss alternative MWCP solvers is very helpful. To better position our greedy solver within the broader combinatorial optimization literature, we briefly compare it with representative exact and local-search approaches below.
> > >
> > > ---
> > > ### Our greedy solver
> > >
> > > Let $b=(1-p)N$ denote the number of selected samples, and let $\bar m$ denote the average neighborhood size in the sparsified graph. We maintain current scores with a max-heap and update only the neighbors of each newly selected sample using reverse adjacency lists.
> > >
> > > The greedy stage has two parts in our practical implementation: (i) heap initialization (to speed up search), which costs $O(N\log N)$, and (ii) $b$ iterative updates. In each iteration,
> > > we extract the current best sample from the heap costs $O(\log N)$, and update the affected neighbors costs $O(\bar m\log N)$, since only the neighbors of the newly selected sample need score updates.
> > > Therefore, each iteration costs:
> > > $$
> > > O(N\log N)+O\big(b(1+\bar m) \log N\big).
> > > $$
> > >
> > > Under our graph sparsification scheme, each sample only interacts with samples in its local cluster, so $\bar m \approx \frac{N}{K} \gg1$, where $K$ is the number of clusters. This gives the following approximation of the above equation:
> > > $$
> > > O\big(N\log N+b(1+\bar m) \log N\big) \approx O(N\log N),
> > > $$
> > > which is close to sorting-like complexity in practice and substantially cheaper than recomputing pairwise gains over all remaining samples at every step.
> > >
> > > ---
> > > ### Branch-and-bound with weighted-coloring bounds
> > >
> > > A exact solver is the branch-and-bound algorithm of _(Babel 1994)_, which uses weighted coloring to compute upper bounds and guide branching. In general, its runtime can be written as $O(BN^2)$, where $B$ is the number of visited branch-and-bound nodes and the $O(N^2)$ term reflects repeated weighted-coloring and pruning operations at each node. Since $B$ is exponential in the worst case, the overall complexity remains exponential.
> > >
> > > In our setting, each subgraph is fully connected, so clique feasibility no longer provides strong pruning, and the problem reduces to selecting a fixed-size subset of size $b=(1-p)N$. This solver therefore essentially searches over size-$b$ subsets. By Stirling’s approximation,
> > > $$
> > > \binom{N}{(1-p)N}\approx e^{N H(1-p)},
> > > $$
> > > where $H(q)=-q\ln q-(1-q)\ln(1-q)$ is the binary entropy function. This is the standard asymptotic approximation for binomial coefficients when the subset size is a fixed fraction of $N$, and therefore gives a natural complexity estimate here. Hence, the worst-case complexity is on the order of $O\left(e^{N H(1-p)}N^2\right)$.
> > >
> > > Even if the same sparsification is applied so that each cluster has size about $N/K$, the complexity only reduces to approximately
> > > $$
> > > O\left(\frac{N^2}{K}\,e^{\frac{N}{K}H(1-p)}\right),
> > > $$
> > > which still remains exponential.
> > >
> > > ---
> > > ### Heuristic local search
> > >
> > > We also analysis the complexity of a heuristic phased local search method _(Pullan, 2008)_, which repeatedly $T$ times of  _add/swap/perturb_ op. on the current clique. In our fixed-size setting, a natural budget is $T=O(b)$, which gives an overall complexity of about $O(Nb^2)=O((1-p)^2N^3)$, i.e., roughly $O(N^3)$.
> > >
> > > ---
> > > For clarity, we summarize the three methods below:
> > >
> > > |Method|Type|Complexity in our setting|
> > > |-|-|-|
> > > |Ours|Greedy|$O\left(N\log N\right)$|
> > > |_(Babel, 1994)_|Branch-and-bound|$O\left(e^{N H(1-p)}N^2\right)$|
> > > |_(Pullan, 2008)_|Heuristic local-search|$O(N^3)$ under $T=O(b)$|
> > >
> > > These comparisons suggest that exact branch-and-bound methods remain exponentially expensive, and local-search heuristics also introduce substantially higher-order cost. By contrast, our method remains scalable while preserving the unified objective framework.
> > >
> > > ---
> > > **Citations.**
> > >
> > > _(Babel, 1994)_: Babel, Luitpold. "A fast algorithm for the maximum weight clique problem." Computing (1994)
> > >
> > > _(Pullan, 2008)_: Pullan, Wayne. "Approximating the maximum vertex/edge weighted clique using local search." Journal of Heuristics (2008)

---

### Decision · Program_Chairs · 2026-04-30

**Decision:**

Accept (regular)

**Comment:**

Most reviewers were positive about the work. They appreciated that the approach is versatile and general, and contributes a theoretical framework that could be used in future work. The experiments are comprehensive, and the method leads to meaningful wall-clock savings. The paper is moreover well-written and clean. It makes for a good contribution to ICML.

Weaknesses were also reported (note: the corresponding text in this metareview is longer than that for strengths, but this should not be conflated with their importance):
- Some reviewers see the computational overhead, which is larger than the baselines (though likely not very significant compared to the full pretraining time), as a weakness of the work.
- It would be good to bolster the comparison to prior work (especially CRAIG) as well as to alternative MWCP approaches, as discussed with reviewer aeuM, whose raised score is based on such assumed improvement. Please also add the complexity comparisons from the replies to reviewer QSiU.
- The graph-based framing of the problem as a "weighted maximum clique" has been argued to be slightly misleading and to overstate the approximation difficulty, which is that of submodular maximization.
- Performance margins over strong baselines are arguably somewhat small.